



# Secondary Aerosol Formation in Incense Burning Particles by Ozonolysis and Photochemical Oxidation

Zhancong Liang[1,2], Liyuan Zhou[1,2], Xinyue Li[1], Rosemarie Ann Infante Cuevas[1,2], Rongzhi Tang[1,2], Mei Li[3,4], Chunlei Cheng[3,4], Yangxi Chu[5], Chak K. Chan[1,2,6]*

[1] School of Energy and Environment, City University of Hong Kong, Hong Kong, China
[2] City University of Hong Kong Shenzhen Research Institute, Shenzhen, China
[3] Institute of Mass Spectrometry and Atmospheric Environment, Guangdong Provincial Engineering Research Center for On-line Source Apportionment System of Air Pollution, Jinan University, Guangzhou 510632, China
[4] Guangdong-Hongkong-Macau Joint Laboratory of Collaborative Innovation for Environmental Quality, Guangzhou 510632, China
[5] State Key Laboratory of Environmental Criteria and Risk Assessment, Chinese Research Academy of Environmental Sciences, Beijing, 100012, China
[6] Low-Carbon and Climate Impact Research Centre, City University of Hong Kong, Hong Kong, China

*Correspondence to*: Chak K. Chan (*chak.k.chan@cityu.edu.hk*)

**Abstract.** Incense burning is a common religious activity that emits abundant gaseous and particulate pollutants into the atmosphere. During their atmospheric lifetime, these gases and particles are subjected to (photo-)oxidation, leading to the formation of secondary pollutants. We examined the oxidation of incense burning plumes under $O_3$ exposure and dark condition using an oxidation flow reactor connected to a single particle aerosol mass spectrometer (SPAMS). Nitrate formation was observed in incense burning particles, mainly attributable to the ozonolysis of nitrogen-containing organic compounds. With UV on, nitrate formation was significantly enhanced, likely due to $HNO_3/HNO_2/NOx$ uptake triggered by OH chemistry, which is more effective than ozone oxidation. The extent of nitrate formation is insensitive to $O_3$ and OH exposure, which can be explained by the diffusion limitation on interfacial uptake. The OH-aged particles are more oxygenated and functionalized than $O_3$-aged particles. Oxalate and malonate, two typical secondary organic aerosols (SOA), were found in OH-aged particles. Our work reveals that nitrate, accompanied by SOA, can rapidly form in incense-burning particles upon photochemical oxidation in the atmosphere, which could deepen our understanding of air pollution caused by religious activities.

## 1 Introduction

Incense burning is a common religious ritual, especially in Asian and African communities (Ye et al., 2016; Khezri et al., 2015; Sidibe et al., 2022), with a massive amount of particles emitted (Lyu et al., 2021; See and Balasubramanian, 2011). The $PM_{2.5}$ concentration at a shrine area reached 6-8 times higher than usual during the Chinese New Year in Chiang Mai, Thailand, mainly due to incense burning (Bootdee et al., 2016). The particle emission factor (i.e., the mass ratio of the emitted particles



to the total material burnt) from incense burning could be up to 10 times higher than those from burning of various types of
biomasses such as rice straw (Akagi et al., 2011; See and Balasubramanian, 2011; Goel et al., 2020).
Previous research mainly focused on the chemical compositions and potential health impacts of fresh incense particles (Li et
al., 2012; Wang et al., 2006; Chuang et al., 2011; Lee and Wang, 2004). However, it was rarely considered that fresh particles
would also be exposed to other atmospheric pollutants and light, which could initiate chemical reactions. The formation of
secondary particulate pollutants could take place during the atmospheric aging of particles (Hodshire et al., 2019; Kumar et
al., 2018; Rudich et al., 2007). For example, our recent work reveals rapid sulfate formation in fresh incense burning particles
upon $SO_2$ exposure in the dark, and it can be accelerated under light (Liang et al., 2022a). Sulfate formation in incense burning
particles under dark and light was mainly attributed to gaseous oxidants and particulate photosensitizers, respectively.
Ozone and OH radicals are two of the most common oxidants in the atmosphere, contributing to secondary inorganic and
organic aerosol formation (Volkamer et al., 2006; Kroll et al., 2006; Chen et al., 2011; Liu et al., 2019; Liu et al., 2018).
Incense burning plume has been reported to contain various volatile organic compounds (VOCs) and NOx (Lee and Wang,
2004; Ho and Yu, 2002), in addition to particulate pollutants. Their interactions with ozone and OH radicals may lead to
secondary aerosol formation. For instance, oxidations of NOx by ozone and OH radicals were considered primary sources of
particulate nitrate (Seinfeld and Pandis, 2008; Liang et al., 2021; Gen et al., 2022). Nevertheless, there is still a lack of
understanding of the secondary aerosol formation in incense burning particles upon atmospheric aging of the plume, which
could potentially worsen air quality, especially near areas of intense religious activities.
This study examines the secondary aerosol formation in fresh incense burning particles under ozone and OH exposure using a
Gothenburg Potential Aerosol Mass (Go: PAM) flow reactor. We first characterized single fresh incense burning particles,
followed by aged particles, with a single-particle aerosol mass spectrometer (SPAMS). Control experiments were performed
to get insight into the possible secondary aerosol formation pathways and their significance. Then, we discussed nitrate
formation as a function of particle size, ozone, and OH exposure. The fragmentation severely hinders the characterization of
secondary organic aerosol (SOA). Nevertheless, -89[$C_2HO_4$] (oxalate) and -103[$C_3H_3O_4$] (malonate), two commonly
considered SOA(Cheng et al., 2017; Sullivan and Prather, 2007), were found.

## 59 2 Experimental

### 60 2.1 Aging of incense-burning particles

The schematic of the experimental set-up can be found in Figure S1 and our previous publication (Liang et al., 2022a). In brief,
we burnt an incense stick in a 20 L glass burning bottle. The burning was rapidly converted from flaming to smoldering after
ignition. A two-stage system diluted the emissions with an overall dilution of around 1600. Compressed air (~0.1 L min$^{-1}$) was





used to introduce the diluted incense burning particles to the PAM reactor equipped with two UVC light tubes (30W, Philips
TUV, $\lambda$max = 254nm). In the control experiments, a charcoal absorber or HEPA filter was used to remove the gaseous
pollutants or particles prior to the introduction to the PAM. The removal efficiency of NOx, VOCs, and particles are ~85%,
~90%, and ~100%, respectively. A controlled dry-wet mixed carrier flow of compressed air (~4 L min$^{-1}$) and a flow of $O_3$
(~0.1 L min$^{-1}$) generated by passing $O_2$ (99.995%, Linde) to an $O_3$ generator (Model 610, Jelight Company Inc, USA) were
introduced into the PAM. [$O_3$] ranged from 300 to 1500 ppb, equivalent to an atmospheric ozone exposure of 10-50 min,
assuming ambient concentration of 60 ppb (Xia et al., 2021). The relative humidity (RH) at the exit of the PAM was monitored
by an RH sensor (M170, Vaisala, Finland). All the experiments were conducted at 80% RH. The exhaust of the PAM was
characterized by an $O_3$ analyzer (106L, 2B technology, USA), a water-based condensation particle counter (WCPC, Aerosol
dynamics Inc, USA), and a SPAMS (Hexin Analytical Instrument Co., Ltd, China). The particles were passed through a
diffusion dryer before entering the PAM, to reduce the matrix effects from water (Neubauer et al., 1998). We also collected
particles on 47 mm quartz filters (PALL, USA) at the exhaust of the PAM reactor for offline analysis. The number of particles
collected on the filters was estimated by the total WCPC counts during the sampling period. The filter sample was extracted
by deionized (DI) water for analyzing water-soluble ions (e.g., nitrate, formate, potassium) by Ion chromatography (IC) using
the same protocol reported in our previous work (Liang et al., 2022a).

We studied the aging of the particles under 'UV', '$O_3$ and dark', and '$O_3$ and UV' in the PAM. Since UV at 254 nm is expected
to photolyze $O_3$ to form OH radicals in the presence of water vapor, we named these aged particles UV-aged, $O_3$-aged, and
OH-aged, respectively. Although 254 nm is not atmospherically relevant, UV-aged particles are used as a reference in the
discussions of the properties of OH-aged particles. The OH exposure, equivalent to the product of gas-phase OH concentration
and residence time, was determined by introducing a stream of $SO_2$ to the PAM for consuming OH radicals and monitoring
the [$SO_2$] decay, following a well-established approach in the literature (Kang et al., 2007). [$SO_2$] was almost constant under
UV on but without $O_3$, suggesting that the photochemistry of incense plume does not affect our estimation of OH exposure.
The upper limit of OH exposure used in this study varied from $1 \times 10^{10}$ to $5 \times 10^{10}$ molecules cm$^{-3}$ s, equivalent to 2~10 hours
of photochemical aging, assuming an ambient OH concentration of $1.5 \times 10^6$ molecules cm$^{-3}$ (Mao et al., 2009).

**2.2 SPAMS analysis**
A detailed description of the SPAMS can be found in Li et al (Li et al., 2011). After the particle flow exits the PAM reactor,
it first passes a PM$_{2.5}$ cyclone to avoid clogging before entering the SPAMS through a 0.1 mm critical orifice at 80 mL min$^{-1}$
flow. Particles achieved a terminal velocity in the supersonic expansion airflow and were detected and aerodynamically sized
by two continuous diode Nd: YAG laser beams (532 nm). They were then ionized by a pulsed Nd: YAG laser (266 nm)
triggered based on the velocity of a specific particle. The positive and negative ions produced were detected according to the
different mass-to-charge ratios (m/z). The energy of the ionization laser was kept at ~0.6 mJ (Cheng et al., 2017). Spectra of





more than 3000 individual particles collected for ~15 min were used for further analysis for each experiment. The instrument
was routinely calibrated with polystyrene latex spheres of 0.2-2.5 μm diameter (Nanosphere Size Standards, Duke Scientific
Corp., Palo Alto). An adaptive resonance theory method (ART-2a) based on MATLAB was used to categorize the incense
particles of similar SPAMS spectral characteristics into different particle groups (Phares et al., 2001). In the ART-2a analysis,
we used a vigilance factor of 0.85, a widely adopted high level (Xu et al., 2018; Wang et al., 2019), and more than 98% of the
particles were analyzed.

## 3 Results and discussions

### 3.1 Single-particle characteristics of incense burning particles

The relative peak area (RPA), defined as the peak area of a specific peak divided by the total positive or negative mass spectral
peak area, can reflect the relative abundance of particulate components (Liang et al., 2022a). The average spectra of the incense
burning particles (Figure 1a) are similar to our previous work on incense burning at 50% RH (Liang et al., 2022b). $+39[K]$
dominates the positive spectra, and organic nitrogen (ON) peaks (i.e., $-26[CN]$ and $-42[CNO]$) from nitrogen-containing
organics (NOC) dominate the negative spectra (Zhang et al., 2020; Zhai et al., 2015; Zhang et al., 2021). These features are
also found in biomass burning particles (Bi et al., 2011; Peng et al., 2019; Luo et al., 2020).

ART-2a categorizes fresh incense burning particles into K-ON, K-ONEC, K-Cl, and OC-ON. Briefly, the "K" and "OC" before
the hyphen indicate the characteristics of the positive spectra, while "ON", "ONEC" and "Cl" after the hyphen indicate the
features of the negative spectra. "K-" particles contain a dominant $+39$ peak and a small $+41$ peak attributed to isotopic
potassium (Bi et al., 2011). On the other hand, the "OC-" particles are rich in typical organic fragments such as $+27[C_2H_3]$
(Cheng et al., 2017). According to the negative spectra, "-ON" particles have dominant ON signals. "-ONEC" particles have
elemental carbon (EC) peaks of $-12n[C_n^-]$, with intensities comparable to typical ON peaks (Zhou et al., 2020). "-Cl" particles
have prominent $Cl^-$ ($m/z=-35, -37$(isotopic)) and $KCl_2^-$ ($m/z=-109, -111$(isotopic)) peaks (Bi et al., 2011). The average spectra
of each category can be found in Figure S2. There are slightly fewer K-ON particles and more K-ONEC particles observed at
80% RH (this work) than at 50% RH in Liang et al (Liang et al., 2022b), probably due to the lower organic concentrations at
higher RH to limit particle-phase partitioning of volatile organic compounds (Donaldson and Vaida, 2006; Mcfall et al., 2020;
Chan and Chan, 2011; Chan et al., 2010). Overall, the number fraction (NF) of each category is similar to our previous work,
with a descending order of K-ON ($47.3\pm5.2\%$) $\gg$ OC-ON ($25.7\pm4.7\%$) $\approx$ K-ONEC ($20.2\pm2.8\%$) > K-Cl ($5.1\pm1.1\%$) (Figure
1c), reflecting the fresh incense burning particles are organic-rich (Li et al., 2012; Zhang et al., 2022a).





**3.2 Ozonolysis of the incense burning particles**

Figure 1a also shows the average spectra of aged incense burning particles under 800 ppb $O_3$. Qualitatively, the major peaks are similar to those in fresh incense burning particles, except for the rise of -62[$NO_3^-$] and -46[$NO_2^-$], which indicates the formation of nitrate and probably nitrite. The formation of organo-nitrate is not considered significant due to the decreased -26[CN] and -42[CNO].

To compare the changes in the organic signals, we first excluded all inorganics and EC peaks (Table S1). Control experiments atomizing $KNO_3$ solution (as $K^+$ is the main inorganic cation found in incense burning particles) showed the RPA ratio of -16[O] to nitrate peaks is $(6 \pm 1.7)$ % due to fragmentation. Sulfate shows negligible fragment under our experimental conditions (Liang et al., 2022a). Thus, we subtracted the RPA of -16[O] by 6% RPA of nitrate. Then, we recalculated the relative peak area (RPA) of the organic peaks only. Figure 1b shows the differences in the spectra of the aged and fresh particles. The positive difference spectra show an RPA increase in the hydrocarbon +37[$C_3H$] but an RPA decrease in +51[$C_4H_3$](Dall'osto et al., 2013). Besides, the increase of +28[CO], +42[$C_2H_2O$], and +43[$C_2H_3O$] indicates the formation of oxidized organics in the particles during ozonolysis. The negative difference spectra show a decrease in ON peaks, possibly due to the destruction of C-N bonds under ozonolysis, and an increase of -45[$CHO_2$] formate peak.

$NO_2$ emitted by incense burning may hydrolyze in the incense burning particles to form nitrite and nitrate ($2NO_2 + H_2O =>$ $HNO_2 + HNO_3$) (Finlayson-Pitts et al., 2003; Ramazan et al., 2006). The uptake of $NO_2$ is slow in deionized water ($\gamma \approx 10^{-7}$), but it could be significantly promoted by chloride ($\gamma \approx 10^{-3}$ in 1mM NaCl) (Enami et al., 2009; Yabushita et al., 2009), which is found as major inorganic anion in the incense burning particles. In addition, the reaction between $NO_2$ and $O_3$ produces $NO_3$ radicals, which could react with organics to form organo-nitrate (Ng et al., 2017). However, control experiments using a charcoal absorber to remove NOx only show ~20% decrease in RPA of total nitrate in $O_3$ aged particles (Figure S3), indicating that $NO_2$ hydrolysis and nitration may not be the main contributor to the nitrate formation. We categorized the $O_3$-aged particles into 7 groups of K-ON, K-ONEC, K-ONN, K-N, K-Cl, OC-N, and OC-ON particles by ART-2a. The definitions of K-, OC-, -ON, -ONEC, -Cl are the same as before. -N particles show prominent nitrate peaks (-46[$NO_2$] and -62[$NO_3$]) in the negative spectra, while -ONN particles show comparable ON peaks and nitrate peaks (Figure S2). The NF of different categories descends in the order of K-ON (29.0±0.7%) ≈ K-ONEC (22.8±1.8%) ≈ OC-N (21.9±1.0%) > K-ONN (11.2±0.9%) > OC-ON (7.6±1.7%) > K-Cl (3.9±0.4%) > K-N (2.5±0.5%) (Figure 1c). Interestingly, the K-ONEC and K-Cl NFs are similar before and after aging, whereas the K-ON NF decreased, and the decrease is comparable to the sum of the K-ONN and K-N NF increases. OC-ON was the only fresh OC- particle type, but OC-N was dominant after aging. A control experiment with a HEPA filter before the PAM showed no detectable particles by SPAMS. Thus, we assume the total SPAMS-detectable particle number was constant before and after aging, and $O_3$-aging may have preferentially converted some -ON type particles to nitrate-containing particles (i.e., -ONN, -N). Besides, $O_3$-aged particles have lower ON (i.e., the sum of -26[CN] and -





42[CNO]) absolute peak area (APA) and higher total nitrate APA (i.e., -46[NO$_2$] and -62[NO$_3$]), than fresh particles (Figure
S4). We considered -46[NO$_2$] mainly as a nitrate fragment but not nitrite since the IC-measured [NO$_2^-$]/[NO$_3^-$] in the water-
extract of collected particles was very low (~0.01) (Figure S5). We used APA here because it reflects the total abundance of
ions and would not affect the analysis of other ions (Spencer and Prather, 2006). The formation of nitrate would increase the
total peak area and decrease the RPA of other peaks, even the APA of others kept constant. Figure S6 also shows a positive
correlation between the total nitrate RPA and formate RPA in the aged particles. Based on offline IC analysis, the water-extract
of O$_3$-aged particles has higher [Formate]/[K$^+$] and [NO$_3^-$]/[K$^+$] in than fresh particles (Figure S7), assuming that K$^+$ is not
reactive and used as an internal standard (Figure S8). Taking these altogether, nitrate and formate likely formed together,
preferentially on K-ON and OC-ON particles. Ozonolysis of NOC has been reported to generate nitrate and formate (Sharma
and Graham, 2010; Yao et al., 2020).

### 3.3 Photochemical oxidation of incense burning particles

With UV (254nm) on, the 800 ppb O$_3$ was partly photolyzed to generate OH radicals in the presence of water vapor, resulting
in an OH exposure of ~3× 10$^{10}$ molecules cm$^{-3}$ s, equivalent to a photochemical age of ~5 h. We will use xx ppb O$_3$ (initial
concentration) +UV, instead of OH exposure, to describe OH aging. The average spectra of OH-aged particles are generally
similar to that of O$_3$-aged particles, with potassium and nitrate peaks dominating the positive and negative spectra, respectively
(Figure 1a). However, the RPA of -46[NO$_2$] and -62[NO$_3$] were 0.2 and 0.4, around 2 times higher than O$_3$-aged particles,
likely indicating more nitrate formation. As will be discussed later, photochemistry triggered by light-absorbing compounds
such as photosensitizers and Fe salts is a possible source of nitrate formation in OH-aged particles.[51-53] However, its
contribution is considered minor compared with OH chemistry since UV-aged particles only show a total nitrate RPA of 0.05,
much lower than that of OH-aged particles (~ 0.7, will be discussed later). Control experiments using a charcoal absorber to
remove the NOx significantly reduced the RPA of total nitrate by ~75% (Figure S3). These suggest that OH chemistry
involving NOx dominated the particulate nitrate formation under OH exposure. Under 800 ppb O$_3$ and UV, the ~90% reduction
of [NOx] with a simultaneous increase in total nitrate peaks under UV suggests the oxidation of NOx by OH radicals to form
HNO$_2$ and HNO$_3$, which can be uptake by the particles afterward (Finlayson-Pitts and Pitts Jr, 1999). Reactive uptake of NOx
initiated by OH chemistry cannot be excluded.

Similar to the O$_3$-aged particles, OH-aged particles show decreases in ON and other organic peaks (+38[C$_3$H$_2$], +50[C$_4$H$_2$],
and +51[C$_4$H$_3$]) in the difference organic averaged spectra (Silva and Prather, 2000), likely due to oxidative consumption by
OH radicals (Figure 1b). The ON peaks decrease in OH-aged particles was more significant than in O$_3$-aged particles, whereas
the increase in formate peak is less obvious. These indicate that NOCs can also be effectively degraded via OH oxidation.
Using the commonly used general markers of oxidized/aged organics in single-particle mass spectrometric studies of -16[O],
-17[OH], +42[C$_2$H$_2$O], and +43[C$_2$H$_3$O] as examples (Taiwo et al., 2014; Denkenberger et al., 2007; Qin and Prather, 2006),





the RPA increase in OH-aged particles are 18, 10, 3, and 17 times higher than in $O_3$-aged particles. This suggests that OH
aging produced more oxidized and functionalized products than $O_3$ aging. The difference average organic spectra of UV-aged
particles almost showed no noticeable peaks, indicating that the chemistry initiated by particulate photoactive compounds may
not be essential to the transformation of the organics (Figure S9).

The OH-aged particles can be categorized into K-ONN, K-N, and OC-N, and they generally have more intense nitrate peaks
than $O_3$-aged particles. Still, "-ONN" particles have comparable ON and nitrate peaks, and "-N" particles have dominant nitrate
peaks in the negative spectra (Figure S2). The NF descends in the order of OC-N (35.7±7.2%) ≈ K-N (35.5±4.2%) > K-ONN
(25.7±2.1%) (Figure 1c). Notably, the NF of OC- particles of OH-aged particles is 50% larger than the fresh particles, likely
due to the formation of additional particulate organics. We could not identify any preferential nitrate formation in specific
particle types since most of the particles have high RPA of nitrate.

**205  3.4 The formation of secondary nitrate**

Figure 2a shows the RPA of nitrate peaks under UV and different exposure of $O_3$ and OH. Since fresh particles also have high
NF of total nitrate, NF cannot accurately depict the effectiveness of nitrate formation. Fresh incense burning particles exhibit
very low RPA of total nitrate, whereas exposure to $O_3$ increases the RPA from almost 0 to around 0.2, irrespective of the [$O_3$].
Only a slight increase (~0.02) in total nitrate RPA was observed for UV-aged particles. However, with both $O_3$ and UV on,
the RPAs of total nitrate further increased to above 0.7, which is also independent of the initial [$O_3$]. Consistent with the
average spectra shown before, nitrate formation due to OH oxidation is likely more efficient than that by ozonolysis. Under
both $O_3$ and OH exposure, the summed APA of nitrate peaks increased as particle size increased, suggesting possibly a larger
total amount of nitrate formed in larger particles (Figure 2b, d). However, the RPA shows an opposite trend, which can be
interpreted as lower nitrate concentration in larger particles. Larger particles have larger surfaces but smaller surface-to-volume
ratios, which lead to a larger absolute amount of nitrate formed but a lower relative concentration of particulate nitrate (Figure
2c, e). Under $O_3$+UV, it is also possible that comparable $HNO_3$ was generated under excess [OH] and contributed to the similar
total nitrate RPA since the [NOx] reductions under different OH exposure are similarly high (Figure S10). The insensitivity of
nitrate formation to $O_3$ and OH exposure can be potentially explained by the diffusion limitation of interfacial uptake due to
the poor hygroscopicity of fresh incense burning particles (Li and Hopke, 1993; Zaveri et al., 2018; Slade and Knopf, 2014;
Liang et al., 2022a).

**222  3.5 The Potential formation of SOA**

Oxalate and malonate are two major dicarboxylates in atmospheric particles and are considered SOA (Yao et al., 2002). They
have been widely studied using single-particle mass spectrometry with well-validated detection efficiency, without severe





complications in mass spectra due to fragmentations (Cheng et al., 2017; Sullivan and Prather, 2007). Figure 3a shows the NF
ratio (aged particles to fresh particles) of oxalate and malonate. We used the NF ratio rather than the APA or RPA, to avoid
large uncertainties in organic abundance due to the much weaker peaks of organics in the spectra.

NFs of malonate and oxalate increase with OH exposure, to 30 and 9 folds, respectively, at 1500 ppb $O_3$ and UV. This trend
is opposite to the independence of nitrate formation on OH exposure, probably because their formations are relatively slower.
These are lower estimates due to the possible degradation by photolysis of Fe-decarboxylate complexes to $CO_2$ (Gen et al.,
2021). In contrast, no oxalate and malonate were observed during ozonolysis, irrespective of $[O_3]$. Furthermore, UV-aged
particles did not show an NF increase of both, indicating that the oxalate and malonate formation were mainly due to OH
radicals, rather than oxidants from particulate photoactive compounds or ozone. The control experiment with a charcoal
absorber shows around 60% and 70% NF reduction of oxalate and malonate, suggesting that the precursors are mainly in the
gas phase (Figure S11). The size distribution of oxalate and malonate containing particles skewed towards the larger sizes,
supporting their nature of secondary formation (i.e., oxidized gaseous precursors were added to the particles that cause size
increase, Figure S12). Figure 3b shows the NF of oxalate and malonate in different categories of the particles. The particles in
the replicated experiments under the same OH exposure were combined to compensate for the low particle concentrations.
The error bars show one standard deviation among different OH exposures. In descending order, the NF of oxalate and
malonate was K-N > K-ONN > OC-N. The mass hygroscopic grow factor (i.e., the mass ratio of wet particles to dry particles)
of inorganic potassium salts $KNO_3$ and KCl at 80% RH are around 1.6 and 2.2 based on AIOMFAC model predictions (Text
S1, https://aiomfac.lab.mcgill.ca/about.html (Zuend et al., 2008)), much higher than that in the water extract of biomass
burning particles (1.1-1.4, including both lab-generated and ambient collected) (Rissler et al., 2006; Carrico et al., 2008; Chan
et al., 2005), as well as fresh incense burning particles (around 1) (Liang et al., 2022b), which are organic-rich. The likely
higher fraction of hygroscopic inorganic of inorganic fraction allows K-N and K-ONN to retain more liquid water to dissolve
gaseous SOA precursors for oxalate and malonate formation. The difference of oxalate and malonate NF is statistically
significantly different between K-N and OC-N (i.e., P <0.05 in ANOVA test). There are many other changes in the NF of
organic fragments, which suggest the oxidation of primary organics and the formation of SOA. However, further analysis was
limited by the lack of molecular information after severe fragmentation. The major spectral evolution and possible peak
attribution can be found in Text S2.

## 4 Conclusions

In this work, we report single-particle characteristics of incense burning particles upon ozonolysis and photochemical
oxidation. Nitrate formation initiated by $O_3$ is generally considered to involve the so-called $N_2O_5$ pathway, in which oxidation
of NOx forms $NO_3$ radical and then $N_2O_5$, which hydrolyzes to form particulate nitrate (Zhao et al., 2021; Xiao et al., 2020).
In our study, nitrate formation was found, as indicated by the increase of total nitrate RPA from near 0 to around 0.2, upon $O_3$





exposure. We propose that ozonolysis of NOCs may be a potential pathway for nitrate formation, in addition to the $N_2O_5$
pathway. With UV on, ozone was photolyzed to form OH radicals, and we observed a significant increase in total nitrate RPA
to above 0.7 at 300 ppb $O_3$ or above. Nitrate formation in OH-aged particles is more prominent than in $O_3$-aged particles and
is attributed to multiphase OH oxidation involving NOx, such as $HNO_3$/$HNO_2$/NOx uptake (Chen et al., 2020; Lu et al., 2019).
At 300 ppb $O_3$ and UV in this study, the equivalent OH and $O_3$ exposure time of the incense particles is estimated to be ~2 h
and ~10 min, respectively, assuming daytime OH and $O_3$ concentration of $1.5 \times 10^6$ molecules cm$^{-3}$ and 60 ppb (Xia et al.,
2021; Mao et al., 2009). Despite the differences in the estimated exposure time for OH and $O_3$, nitrate formation in incense
particles under sunlight can be efficient.

We also observed various changes in organics peaks, though less apparent than nitrate in the average spectra. Overall,
oxygenated fragments like +42[$C_2H_2O$] increase, which indicates functionalization of the organics upon oxidation. The
increase of such oxygenated fragments is more significant in OH-aged than $O_3$-aged particles. -26[CN] and -42[CNO]
attributed to NOC decreased under $O_3$ and OH exposure. Apparent formate formation was observed in $O_3$-aged particles, likely
from the degradation of NOC. Production of formate in OH-aged particles was less significant than that in $O_3$-aged particles,
attributed to the photolysis of $O_3$. Oxalate and malonate were observed in OH-aged particles but not in $O_3$-aged particles, and
the NFs increased with OH exposure. Furthermore, oxalate and malonate preferentially formed on K-N particles, followed by
K-ONN and then OC-N, indicating a potentially crucial role of aerosol liquid water in SOA formation.

Though the molecular characterization of SOA is beyond the focus of this work, the formation of oxalate and malonate shed
light on the SOA formation upon photochemical oxidation of the incense burning plumes. Formate and dicarboxylates are
important hygroscopic organics in atmospheric particles, which can potentially act as cloud condensation nuclei (Yao et al.,
2002; Peng and Chan, 2001). Incense burning particles were often used as biomass burning particle surrogates (Li et al., 2012;
Schurman et al., 2017; Zhang et al., 2014; Kuwata and Lee, 2017), due to their similar physicochemical properties and overall
composition (Li et al., 2012; Zhang et al., 2022b). Our work sheds light on the secondary aerosol formation in biomass burning
particles upon exposure of atmospheric oxidants, despite that the detailed composition of incense burning plume may be
different from biomass burning plume because of the manufacturing process of incense sticks. For instance, incense burning
sticks may contain additives such as adhesives beyond biomass constituents (Lin et al., 2007). Future works are encouraged to
explore the formation mechanism and kinetics of secondary pollutants in the incense burning and biomass burning particles.
Due to the short residence time in a PAM reactor, assumption in the interchangeability of oxidant concentration and reaction
time in estimating total exposure was made. However, Chu et al. (2019) challenged such interchangeability in ozonolysis
reaction of linoleic acid (Chu et al., 2019). Aging at ambient concentrations of oxidants should also be investigated.




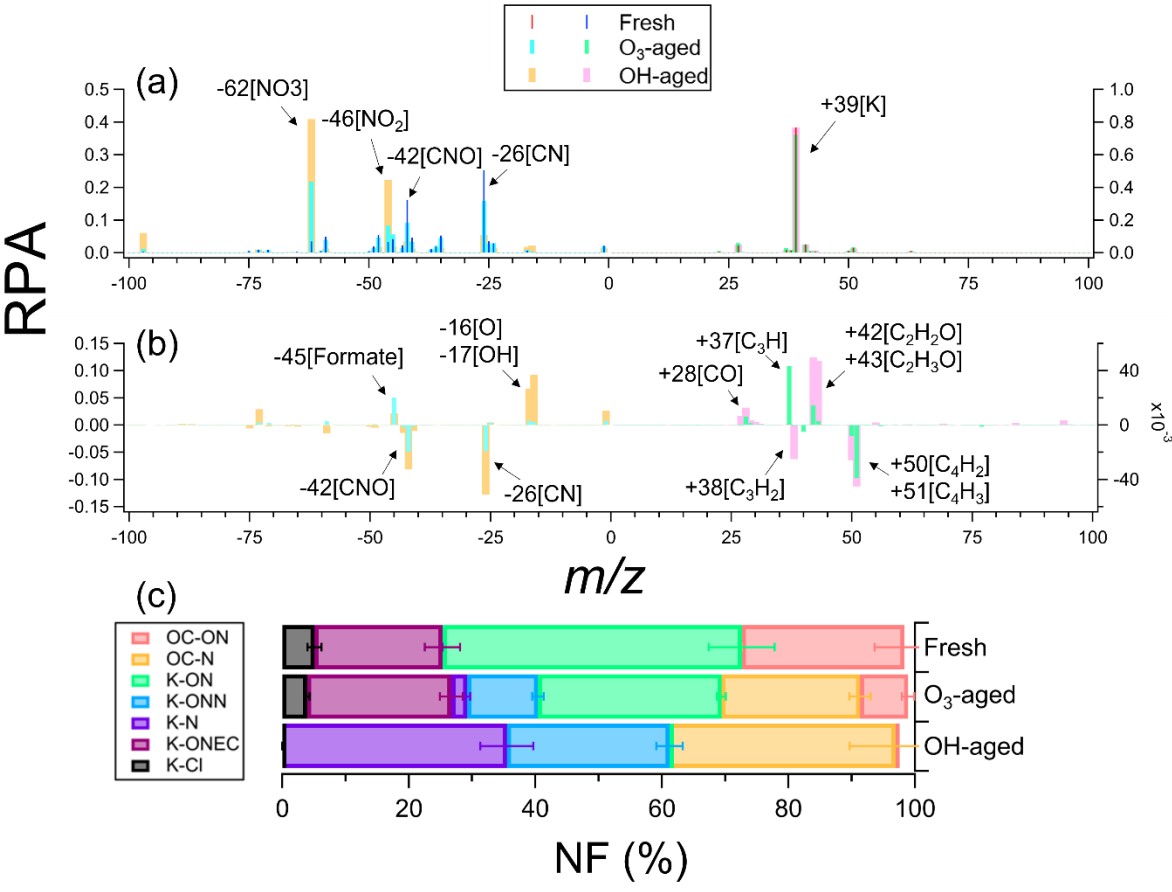

**Figure 1.** (a) The average spectra of fresh and aged incense burning particles at 800 ppb $O_3$(+UV); (b) The difference (aged minus fresh) of the average organic spectra of incense burning particles at 800 ppb $O_3$(+UV). The left axis and right axis are for negative spectra and positive spectra, respectively. (c) Number fraction of different categories in fresh, $O_3$-aged, and OH-aged particles.



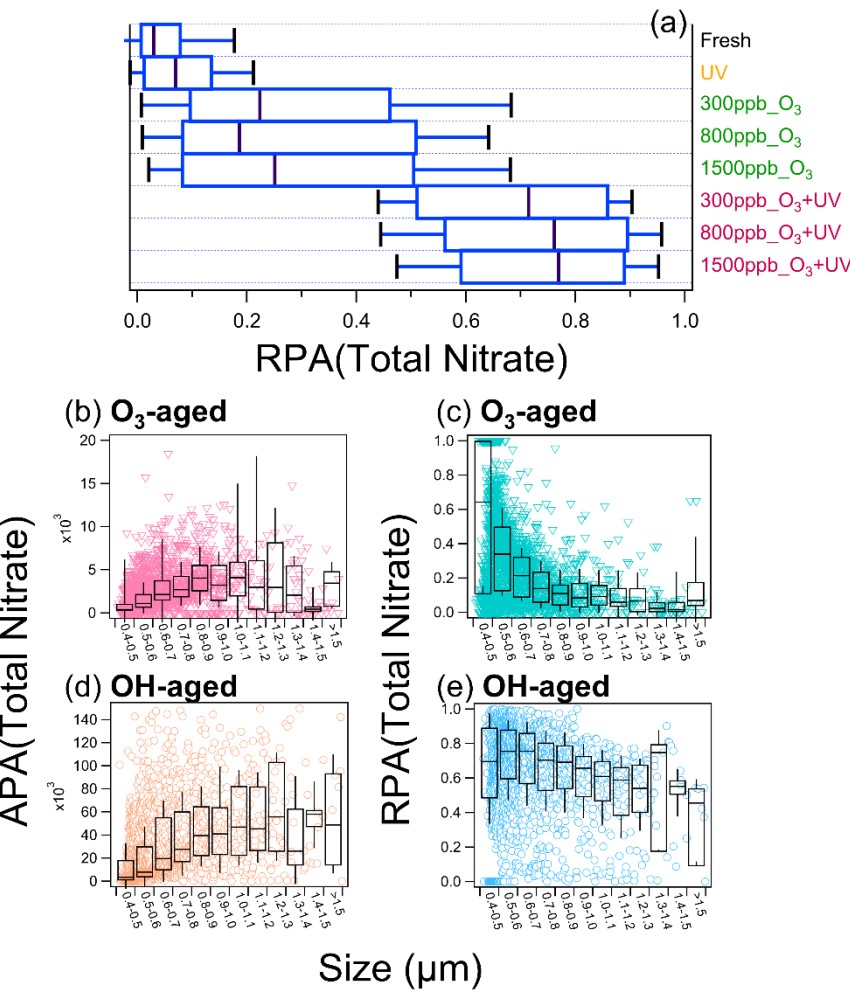

**Figure 2.** (a) The whisker-box plot of total nitrate RPA of fresh and aged particles. The whisker-box plots of (b, d) APA and (c, e) RPA of total nitrate in $O_3$- and OH-aged particles as a function of size (unit: μm) of particles aged at 1500 ppb $O_3$ (+ UV). The medians are shown as the lines in the boxes, and the error bars represent one standard derivation.





309

310

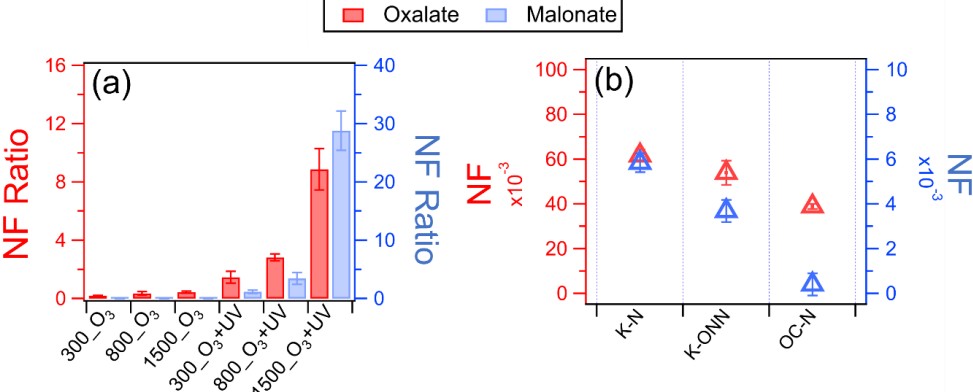

311

**Figure 3.** (a) The NF ratios of oxalate and malonate under different conditions. 300_$O_3$ denotes aging at 300 ppb $O_3$; (b) The

NF of oxalate (left axis) and malonate (right axis) in different categories of aged particles.

*Data availability.* The supplement provides additional figures and tables.

*Competing interests.* The contact author has declared that neither they nor their co-authors have any competing interests.

Financial support.

*Acknowledgment.* We gratefully acknowledge the support from the Key-Area Research and Development Program of

Guangdong Province (2020B1111360001), the Hong Kong Research Grants Council (No.11304121, R1016-20F), the National

Natural Science Foundation of China (No. 42275104, 41905122).

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
