# Peer review of "Secondary Aerosol Formation in Incense Burning Particles by Ozonolysis and Photochemical Oxidation"

_Atmospheric Chemistry and Physics, 2022_

## Author Comment (AC1)

**Response to the Reviewer 1:**

*General Comments*

*Liang et al. (manuscript) describes the experimental formation of SOA and nitrate through chemical aging induced in an oxidation flow reactor (OFR). The authors injected combustion air from incense burning into the OFR at high RH, which were rapidly aged by controlling UV light and O3 to mimic UV-aged, O3-aged, and OH-aged scenarios. The authors used a single-particle AMS (SPAMS) to obtain the chemical composition of the particles and the Gothenburg PAM OFR (Go:PAM) as the reaction vessel.*

*The authors use the adaptive resonance theory method (ART-2a) algorithm to perform cluster/categorization analysis with the mass spectra (Zhao et al., 2008). The authors conclude that the OH-aged case generates more secondary nitrate than the O3-aged case based on the higher relative peak area (RPA). The enhanced secondary nitrate formation is attributed to higher uptake of nitrogen-containing species.*

*The manuscript overall lacks quantitative information, and I am confused about the OFR configuration. The authors provide some [NOx] information in Figure S10 and in the text, but a NOx instrument is not shown in the OFR set up in Figure S1. Moreover, the flow rates entering and exiting the OFR in Figure S1 already match, so adding a NOx monitor would cause a flow imbalance.*

*Moreover, the methodology and instrument details are lacking for replication, and additional explanations are needed to connect the chemistry in these OFR conditions to those of the atmosphere. For instance, the manuscript is missing the Go:PAM temperature and experiment residence times. The generated particle number concentrations from the WCPC, the particle mass collected on the filters for IC analysis, and the mass of incense used are not available. I do not have a clear picture of how much aerosol entered and exited the OFR.*

*The experiments and discussions fall within the scope of Atmos. Chem. Phys., and the content is topical to the atmospheric chemistry community. However, the manuscript is currently underprepared for publication, and there are technical issues that need resolution. Given the importance of understanding how chemical aging impacts aerosol evolution, I request the authors perform major revisions and resubmit the manuscript.*

**Author's Response:** Thank you for the constructive comments. Many of the comments/questions are addressed by the addition and clarification of experimental conditions and result discussions. These comments have helped improve the manuscript and strengthen the analysis of the results, but they do not affect the key conclusions in the manuscript.

SPAMS cannot give quantitative mass concentration information of the aerosol samples, especially the organics. The Aerodyne AMS is generally accepted as quantitative. However, the ionization efficiency is often assumed constant, irrespective of the chemical compositions. Hence, we feel that the inability of accurate mass concentration determination is not necessarily a unique shortcoming of SPAMS. Rather, it has less exposure in the scientific community to yield an
accepted ionization efficiency for all its analytes. Nevertheless, it is a unique and useful tool to
clearly identify the mixing states of particles, which is the emphasis of this paper. For instance, it
clearly shows that nitrate preferentially formed in -ON type particles, and dicarboxylates formation
in K- type particles prevailed over OC- type particles, though the detailed speciation analysis of
the organic compounds is limited by fragmentation. We clarified this limitation of the technique
in the manuscript.

We apologized for the confusion from the Go:PAM schematic (Figure S1). It has been corrected
to provide additional experimental information.  The characterization of the Go:PAM was reported
in Watne et al., 2018 and this system has been used in a number of publications in aerosol aging
(Le Breton et al., 2019; Watne et al., 2018; Wang et al., 2022; Yu et al., 2022; Zhang et al., 2020;
Li et al., 2019; Liu et al., 2019; Xu et al., 2022; Zhou et al., 2021a; Zhou et al., 2021b; Tsiligiannis
et al., 2019).

[Figure]

**Figure S1.** The schematic of the experimental set-up. The NOx analyzer and the VOC sensor were used only in the experiment for determining the gas removal efficiency and NOx decay under OH exposure.

We have also added the temperature, residence time, and WCPC particle counts to the methodology description as shown below. The particles were collected on filters for IC analysis, and the mass of incense used was useful to estimate the emission factor (EF) of incense burning particles, although mass concentration was not the focus of this manuscript.

Line 77-78: The RH and the temperature at the exit of the Go:PAM were monitored by an RH and temperature sensor (M170, Vaisala, Finland). All the experiments were conducted at $80 \pm 0.6$ % RH and $22 \pm 1.7$ °C. The residence time in the Go:PAM was ~100 s.

Line 86-89: We also collected particles on 47 mm quartz filters (PALL, USA) at the exhaust of the Go:PAM reactor for offline mass and chemical analysis. The number of particles collected on the filters was estimated by the total WCPC counts during the sampling period. The particle number concentration from the WCPC was $6100 \pm 2400$ # $cm^{-3}$ and the estimated number of collected particles was around $10^8$ #.

*Specific Comments*

*1. The SPAMS calibration is not outlined, and details on the ART-2a solution is inadequate.*

*Details on the SPAMS operation would help assess the data quality. The Aerodyne soot particle AMS requires laser alignments for consistent measurements (Avery et al., 2020); does the SPAMS in this manuscript need a similar calibration? Particle transmission through the aerodynamic lens is size-dependent (Huang et al., 2013). How would size-dependent particle detection influence the data interpretation?*

*The absolute peak area (APA), relative peak area RPA, and number fraction (NF) are frequently invoked in the data interpretation. The authors use APA and RPA as analogues to concentration (or fraction of total aerosol). However, I suspect that depends on how efficiently different species are ionized by the pulsed Nd:YAG laser, and I would like to know if adjustments have been made to the RPA based on the ionization efficiency. What is the ionization efficiency (IE) of species mentioned in the manuscript, and is IE consistent across species?*

*As for the ART-2a solution, Zhao et al. (2008) and Huang et al. (2013) note that there is no general rule for the vigilance factor, and that a comparative approach (like re-grouping or comparing with other clustering algorithms) may be needed. I also note there is no PAH category, despite particulate PAH found in previous incense combustion studies (Ji et al., 2010) and a PAH contribution being found in a similar mass spectrometer with ART-2a (Passig et al., 2022). Can the authors provide more detail and justification for their ART-2a solution?*

**Author's Response:**

The SPAMS we used operates with a principle similar to the ATOFMS, but not Aerodyne SP-AMS. The laser alignment was calibrated by atomizing Polystyrene Latex (PSL) particles with sizes ranging from 200-1000 nm.

The collection efficiency of SPAMS increased with size from ~1% at 200 nm to ~40% at 960 nm (as shown below). Therefore, the secondary aerosol formation in small particles may have been underestimated.

[Figure]

**Figure S13.** Relation between the particle collection efficiency of Polystyrene Latex particles and diameter in SPAMS.

The matrix effect makes the determination of ionization efficiency (IE) difficult for SPAMS, which is also an issue in other single-particle mass spectrometers (e.g., A-ToF-MS, LAAP-MS). Therefore, unlike the aerodyne AMS, SPAMS is not a quantitative instrument, but is unique for single-particle mixing state characterization, which is the focus of this work. One should note that the Aerodyne AMS is not free from matrix effect either and hence offline chemical composition measurements were usually conducted for comparison. APA and RPA were used as semi-quantitative indicators of the abundance and concentration of chemical species (Zhou et al., 2021b; Cheng et al., 2018). We have changed the title to 'Secondary Aerosol Formation in Incense Burning Particles by Ozonolysis and Photochemical Oxidation via Single Particle Mixing State Analysis' to emphasize the focus of this paper.

There are various algorithms for clustering single particles, but all are based on spectral similarities (Zhao et al., 2008; Rebotier et al., 2007). ART-2a is the sole algorithm incorporated into the commercial SPAMS system, and it is also the most common one for analyzing single-particle data. Although there is no general rule for the vigilance factor in ART-2a, Zhao et al. (2008) reported that both a relatively small vigilance factor (e.g., 0.5 or 0.6) and a relatively high vigilance factor (e.g., 0.8) show very similar clustering accuracies (± 5%). Nevertheless, we have added the
vigilance factor and further clarifications to the manuscript:

Line 226-229: The collection efficiency of SPAMS increased from ~1% at 200 nm to ~40% at 960
nm (Figure S13). Therefore, secondary aerosol formation in small particles may have been
underestimated. However, this would not affect the conclusion of the results that nitrate formed
in incense burning particles upon $O_3$+Dark and $O_3$+UV aging.

Line 113-116: In the ART-2a analysis, we used a vigilance factor of 0.85, and more than 98% of
the particles were analyzed. Note that there is no general rule for the vigilance factor in ART-2a.
Zhao et al. (2008) reported that both a small vigilance factor (e.g., 0.5 or 0.6) and a relatively high
vigilance factor (e.g., 0.8) show very similar clustering accuracies (± ~5%) (Zhao et al., 2008).

We have searched for the signals of particulate PAHs. Passig et al. (2022) identified PAH-
containing particles by the presence of at least four peaks on the m/z channels 178, 189 (fragment
of alkylated phenanthrenes), 202, 220, 228, and 252. We did not find such peaks in any of our
samples, although they were detected in an AMS study (Ji et al., 2010). We have added this
information to the manuscript:

Line 126-130: Despite that particulate polyaromatic hydrocarbons (PAHs) were found in a
previous incense combustion study (Ji et al., 2010) and a recent study of ambient particles based
on a single-particle mass spectrometer with ART-2a (Passig et al., 2022), none of the fresh incense
burning particles in our experiments contained the PAHs peaks (m/z = 178, 189 (fragment of
alkylated phenanthrenes), 202, 220, 228, and 252). Regardless of the presence of PAHs or not, our
conclusion on nitrate formation does not depend on the detection of specific chemicals such as
PAHs.

*2. OFR characterization and operation details are missing.*

*Offline OHexp calibrations may be inaccurate when OH reactive species suppressed OH.*
*Basically, OH suppression is when the external OH reactivity (extOHR) entering the OFR is high*
*enough to titrate the OH, which results in OH-aging being suppressed. In such scenarios, offline*
*OHexp calibrations become unreliable, possibly by orders of magnitude (see Section 3.1.4 of Peng*
*and Jimenez, 2020). Peng and Jimenez (2020) also notes that OFR254 is susceptible to OH*
*suppression at low O3 injections.*

*Operational information of the OFR would be valuable for replication and should be mentioned*
*in the supplementary. Watne (2018) describes the Go:PAM as being made of quartz; have there*
*been efforts to constrain electrostatic particle wall loss (Cao et al, 2020)? How would gas wall*
*loss (Palm et al., 2016) affect the results reported, or is gas wall loss negligible? What cleaning*
*procedure was taken to minimize carryover effects between experiments?*

**Author's Response:**

Watne et al. suggested that the penetration of the particles is close to 100% for particles larger than 100 nm. Hence the wall loss is negligible for the 0.2-2 um particles that SPAMS measures. Besides, a control experiment measuring the total VOCs at the entrance and the exhaust of the Go:PAM suggested that the gas wall loss was also minor (6 ± 4%). We have added this analysis to the manuscript:

Line 80-83: Watne et al. suggested that the penetration of the particles is close to 100% for particles larger than 100 nm. Hence the wall loss is negligible for the 0.2-2 um particles that SPAMS measures. Besides, a control experiment measuring the total VOCs at the entrance and the exhaust of the Go:PAM suggested that the gas wall loss was also minor (6 ± 4%).

Also, we have added details on the OHexp calibration with $SO_2$. The estimation of extOHR requires VOC analysis and is not available in our study. We have also added this limitation to the text.

**Text S1.** Estimation of the OH exposure.

$SO_2$ was used to calculate the OH exposure in the Go:PAM. The UVC lamps were turned on to warm up for ~30 min and turned off. Then, $O_3$ (300, 800, 1500 ppb) and $SO_2$ (~200 ppb) were introduced to the Go:PAM with the UVC lamps turned off until its initial concentration remained constant at steady-state conditions, which typically took around 5 min. The $[SO_2]$ was recorded as $[SO_2]_{Initial}$. After that, the UVC lamps were turned on until the final $[SO_2]$ stabilized and was recorded as $[SO_2]_{Final}$. The time scale for the stabilization of $[SO_2]$ was around 4 min. The OH exposure at each condition is calculated using Eq. (A1):

$$OH\ exposure = \frac{1}{k_{OH,SO_2}} \times -ln\left(\frac{[SO_2]Final}{[SO_2]Initial}\right) \qquad (A1)$$

where $k_{OH,SO2} = 9 \times 10^{-13}$ cm$^3$ molec$^{-1}$ is the bimolecular rate constant between OH and $SO_2$ (Davis et al., 1979). The equation above is the result of integrating the differential rate equation for $SO_2$ and assuming pseudo-first order kinetics. The estimation of external OH reactivity (i.e., the OH reactivity with VOCs) requires VOC analysis and is not available in our study. Therefore, the OH exposure shown in this study may have been underestimated.

*3. Kinetic modeling may be needed for interpretation.*

*The authors' argument on secondary nitrate formation, either heterogeneously or in the gas phase is limited by the lack of quantification HNO3, HONO, NOx, NOy etc. The difference in condensed nitrate between the O3 and OH-aged cases may be due to differences in HNO3/HONO/NOx uptake as the authors allege. A kinetic calculation showing that the formation of these species under the difference OFR conditions are comparable would be more demonstrative.*

*Moreover, gas-phase organic nitrate formation, either through VOC+NO3 or RO2+NO (Ziemann and Atkinson, 2012) and condensation should be considered. Kinetic modeling may be needed to*

*connect the experimental aging conditions in the Go:PAM to those of the atmosphere (Peng and Jimenez, 2020).*

**Author's Response:** Kinetic modeling will be useful in quantitatively evaluating the multiphase kinetics during aging incense burning particles. However, this is beyond the scope of this manuscript. Hence our data interpretation focuses on the formation of nitrate via mixing state analysis of the resulted particles, but not on the detailed mechanisms. It will require another study to conduct detailed modeling with new experimental data. We understand the limitation of this study that the SPAMS is not quantitative and hence have not attempted to do kinetic modeling.

*Line 30: I recall incense burning is found in other cultures and am unsure if the practice is "especially" common in Asian and African religious rituals. I suggest either providing a reference for that point or removing "especially" in this sentence.*

**Author's Response:** Agree, but the use of "especially" has not excluded other possibilities.

*Line 35: The incense burning references cited here mention that there is variation in the particle emission factor (EF) across incense varieties. How does the particle EF in these experiments compare with those previous works? Were the combustion conditions comparable to those previous works?*

**Author's Response:** The EF of the incense burning particles can vary by the type of incense and burning conditions. The theme of this paper is the aging of incense-burning particles and the formation of the secondary components of particulates, but not the particle EF. Again, our conclusion of nitrate formation in single incense burning particles is not affected by the EF. We have added additional information for the burning conditions:

Line 62-64: The air exchange rate per hour (ACH) is 0.3, comparable to the typical natural ventilation conditions (Lee et al., 2004). The relative humidity (RH) and the temperature inside the burning bottle were $56 \pm 9$ % RH and $22 \pm 2.7$ °C.

*Line 62: There is no information on the incense sticks used, like the manufacturer or composition, and Liang et al. (2022) used several as shown in their Figure S21. What type of incense was used here? How much incense was burned? This information could be valuable for replication studies.*

**Author's Response:** We added additional information on the incense sticks to the manuscript:

Line 61-62: In brief, we burnt an incense stick (Figure S2, Kwok Tin Heung, Hong Kong) in a 20 L glass burning bottle for each experiment.

[Figure]

**Figure S2.** The appearance of the incense sticks.

*Line 64: The methods reference (Liang et al., 2022) states there were four UVA lamps, while here*
*the authors say they used "two UVC light tubes." Please confirm that the Go-PAM set up had*
*changed for this manuscript and specify that in the text. Moreover, what lamps were used? Rowe*
*et al. (2020) found that 185 and 254 nm photon fluxes would vary across manufacturers, which*
*may be re*

**Author's Response:** We apologize for the confusion of referring our previous publication for the
schematic. We have now removed the reference to our previous work. We have added detailed
information on the UVC lamps used in this study including the spectra of the lamp to the
supplementary information.

Line 61: The schematic of the experimental set-up is in Figure S1
.

Line 65-67: Compressed air (~0.1 L min$^{-1}$) was used to introduce the diluted incense burning
particles to the Go:PAM reactor equipped with two UVC light tubes (30W, Philips TUV, λmax =
254nm). The spectrum of the lamp was shown in Supplementary information (Figure S3).

[Figure]

**Figure S3.** The emission spectrum of the UVC lamps.

*Line 65: I am confused on how many experiments this manuscript is describing. I see in Figure 2 that there were 7 involving aging and 1 fresh; were some of these the "control" experiments? From this sentence I expect at least 2 types of controls, with either a charcoal absorber or HEPA filter. Were the control cases then also aged?*

**Author's Response:** The data from the control experiments are available in the supplementary information, but not Figure 2. These control experiments are aged experiments. We have further clarified the availability of these data.

Line 67-70: In the control experiments, a charcoal absorber or HEPA filter was used to remove the gaseous pollutants or particles prior to the introduction to the Go:PAM. The data of these control experiments can be found in the supplementary information (Figure S4, S5). All these control experiments were aged experiments.

*Line 66: How did the authors obtain these removal efficiencies?*

**Author's Response:** We have added the explanation for the removal efficiencies evaluation:

Line 70-73: The removal efficiency of NOx, VOCs, and particles were ~85%, ~90%, and ~100%, respectively, determined by the concentration reduction after applying a HEPA filter or charcoal absorber at the exhaust of the burning bottle, using a NOx analyzer (T200, Teledyne) or a Total VOC analyzer (Yuante) (Figure S1).

*Line 67: Compressed air or zero air? If the air is coming from a compressor, were efforts made to scrub the air of contaminants?*

**Author's Response:** The compressed air was treated by a HEPA filter and a charcoal absorber prior to the experiment system. We have clarified this in the manuscript and the schematic figure.

Line 72-75: A controlled dry-wet mixed carrier flow of compressed air (~4 L min$^{-1}$) and a flow of $O_3$ (~0.1 L min$^{-1}$) generated by passing $O_2$ (99.995%, Linde) to an $O_3$ generator (Model 610, Jelight Company Inc, USA) were introduced into the Go:PAM. The compressed air was treated by a HEPA filter and a charcoal absorber prior to the experiment system.

*Line 74: The methods reference (Liang et al., 2022) does not mention using a diffusion dryer. At ~0.1 LPM, what was the residence time in the dryer and is there an estimate of particle loss in the dryer? Was the dryer effective in removing H2O?*

**Author's Response:** The RH at the exhaust of the diffusion dryer was ~15%. The residence time of the particles in the dryer was estimated to be 5 s and the particle loss was ~4% according to the CPC measurements. We have added this additional information to the manuscript.

Line 83-86: The particles passed through a diffusion dryer before entering the Go:PAM to reduce the matrix effects from water (Neubauer et al., 1998). The RH at the exhaust of the diffusion dryer was ~15%. The residence time of the particles in the dryer was estimated to be 5 s and the particle loss was ~4% according to the CPC measurements.

*Line 76: What were the estimated number of particles collected?*

**Author's Response:** The estimated number of the particle collected has been added.

Line 86-89: We also collected particles on 47 mm quartz filters (PALL, USA) at the exhaust of the Go:PAM reactor for offline mass and chemical analysis. The number of particles collected on the filters was estimated by the total WCPC counts during the sampling period. The particle number concentration from the WCPC was $6100 \pm 2400$ # $cm^{-3}$ and the estimated number of collected particles was around $10^8$ #.

*Line 83: Please provide additional details on the OHexp calibration with SO2, in particular the concentrations of SO2 used and timescales to reach equilibrium. An estimate of extOHR during the experiments should be compared with that of the SO2 calibrations.*

**Author's Response:** We have added details on the OHexp calibration with $SO_2$. The estimation of extOHR requires VOC analysis and is not available in our study. We have added this limitation to the text. As discussed earlier (Line 169-182 in this file), the lack of the extOHR does not affect our conclusion that $O_3$+UVaging by OH leads to more significant nitrate formation than aging by $O_3$ alone in incense burning particles.

*Line 101: See specific comment 1.*

**Author's Response:** Please find our response above.

*Line 113: Explaining the abbreviations would improve readability. For instance, OC and ONEC do not appear prior in the text?*

**Author's Response:** The explanation of the abbreviations has been added to the manuscript.

Line 132-135: ART-2a categorizes fresh incense burning particles into K-ON, K-ONEC, K-Cl, and OC-ON. EC, Cl and OC are abbreviations of elemental carbon, chloride and organic carbon, respectively. Briefly, the "K" and "OC" before the hyphen indicate the characteristics of the positive spectra, while "ON", "ONEC" and "Cl" after the hyphen indicate the characteristics of the negative spectra.

*Line 147-149: How does the charcoal absorber remove NOx without removing VOC? How would the removal of VOC affect the interpretation here?*

**Author's Response:** The charcoal absorber removes both NOx and VOC. However, the VOC contents were not expected to be important in the observed nitrate RPA reduction, as the amount of nitrogen-containing VOCs was minor in the incense burning plume, according to the literature (Manoukian et al., 2013).

Line 166-170: However, control experiments using a charcoal absorber to remove NOx only show ~20% decrease in RPA of total nitrate in $O_3$ aged particles (Figure S4), indicating that $NO_2$ hydrolysis and nitration may not be the main contributor to the nitrate formation. The charcoal absorber removes both NOx and VOC. However, it was not expected to be important in the observed nitrate RPA reduction, as the content of nitrogen-containing VOCs was minor in the incense burning plume, according to the literature (Manoukian et al., 2013).

*Line 156: How would the loss of SVOC/LVOC in the HEPA filter (Shilling, 1997) affect the conclusion of the control experiment?*

**Author's Response:** The addition of the charcoal to remove VOCs at the exhaust only caused ~6 % variation of the NF of -ON type particles and nitrate-containing particles (i.e., -ONN, -N). Hence, we considered the loss of SVOC/LVOC unimportant to our conclusion. We have added this to the manuscript:

Line 177-183: A control experiment with a HEPA filter before the Go:PAM showed no detectable particles by SPAMS. Thus, we assume the total SPAMS-detectable particle number was constant before and after aging, and $O_3$-aging may have preferentially converted some -ON type particles to nitrate-containing particles (i.e., -ONN, -N). It has been reported that the HEPA filter would cause the loss of semi-volatile VOC (SVOC) or less-volatile VOC (LVOC) (Schilling, 1997). However, the addition of the charcoal to remove VOCs at the exhaust only caused ~6 % reduction of the NF of -ON type particles and nitrate-containing particles (i.e., -ONN, -N), suggesting the roles of SVOC and LVOC were minor to our conclusion.

*Line 182: How did the authors arrive at the "~90 % reduction of [NOx]"? Was this a separate test? If so, please add a quick summary of how that test was performed.*

**Author's Response:** Yes, it is a separate test. We have clarified in the caption of the schematic.

**Figure S1.** The schematic of the experimental set-up. The NOx analyzer and the VOC sensor were only used in the experiment determining the gas removal efficiency and NOx decay under OH exposure.

*Line 183: Was a NOx monitor available? If so, please provide the monitor's location in Figure S1 and specify in the text. Also, please explain how the flow rates entering and exiting the Go:PAM would be reconciled.*

**Author's Response:** We have revised the experimental schematic.

*Line 191-193: Do OH and O3 oxidation form similar functional groups? Are those functionalities evenly represented in these general markers? The RPA increase of SOA markers in OH over O3-aging may be skewed if these markers overrepresent one oxidation case over the other.*

**Author's Response:** These marker fragments may not be directly related to the functional groups. Rather, they are just an indicator of the abundance of oxygen in the particulate matters. We have clarified this in the manuscript.

Line 214-217: Using the common general markers of oxidized/aged organics in single-particle mass spectrometric studies of -16[O], -17[OH], +42[$C_2H_2O$], and +43[$C_2H_3O$] as examples (Taiwo et al., 2014; Denkenberger et al., 2007; Qin et al., 2006), their RPA increase in $O_3$-UV-aged particles are 18, 10, 3, and 17 times higher than in $O_3$-aged particles. This significant enhancement of RPA suggests that OH aging produced more oxidized  products than $O_3$ aging.

*Line 214: Do larger and smaller particles have similar surface properties, at least with regards to nitrate uptake?*

**Author's Response:** It has been reported that larger organic and inorganic mixed particles could undergo liquid-liquid phase separation (LLPS) more easily than smaller ones (Kucinski et al., 2019). The organics would mainly locate at the outer layers of the particles, whereas the inorganic components reside as a core. The hydrophobic organic shell may have retarded the uptake of $HNO_3$/$HNO_2$/NOx to form nitrate. Profiling the surface composition and properties of the particles is beyond the scope of this paper. We have added this possibility to the manuscript for future works.

Line 241-247: However, the RPA shows an opposite trend, which can be interpreted as lower nitrate concentration in larger particles. Larger particles have larger surfaces but smaller surface-to-volume ratios, which lead to a larger absolute amount of nitrate formed but a lower relative concentration of particulate nitrate (Figure 2c, e). It has been reported that the larger organic and inorganic mixed particles could undergo liquid-liquid phase separation (LLPS) more easily than smaller ones (Kucinski et al., 2019). It is possible that the organics are mainly located at the outer layers of the particles, whereas the inorganic components reside in the core. The hydrophobic organic shell may have retarded the uptake of $HNO_3$/$HNO_2$/NOx to form nitrate.

*Line 222: I suspect SOA formation is not "potential," but rather inevitable under the aforementioned OFR conditions, so I suggest removing "potential" from the section heading. Also, are oxalate and malonate universal and proportional indicators of SOA? That is, do different SOA*

*precursors form these indicators evenly under different oxidants (O3/OH)? I am concerned that*
*there are specific chemical conditions where these species are enhanced without a proportional*
*enhancement of SOA, which may skew the NF.*

**Author's Response:** We have spent a lot of time interpreting the question. It is likely the SOA characteristics and precursors are different under $O_3$ and OH. Hence, we are conservative and use the term "potential", even though oxalate and malonate are commonly found highly oxidized SOA. The total organic content (TOC) ratio of aged to fresh particles extract, which indicates the formation of SOA, was higher upon OH oxidation than $O_3$ oxidation (Figure S16). We have added clarifications to the manuscript:

[Figure]

**Figure S16.** The total organic content (TOC) ratios of aged to fresh particles extract.

Line 89-92: The filter sample was extracted by deionized (DI) water for analyzing water-soluble ions (e.g., nitrate, formate, potassium) by Ion chromatography (IC) using the same protocol reported in our previous work (Liang et al., 2022). The total organic content (TOC) of the water-extract was analyzed by a TOC analyzer (Shimadzu TOC-L).

Line 268-271: In contrast, no oxalate and malonate were observed during ozonolysis, irrespective of $[O_3]$. It is likely the SOA characteristics and precursors are different under $O_3$ and OH. Hence, we are conservative and use the term "potential", even though oxalate and malonate are commonly found highly oxidized SOA. The total organic content (TOC) ratio of aged to fresh particles extract was higher upon OH oxidation than $O_3$ oxidation (Figure S16).

*Line 229: Is NF of oxalate and malonate proportional with SOA concentrations? As it stands, "30*
*and 9 folds" increases of these tracers sounds like SOA increased by that much.*

**Author's Response:** While the TOC ratio at different $O_3$+UV, which indicates the formation of SOA, shows an overall trend similar to the NF ratios of oxalate and malonate (Figure S15), the TOC ratio was 1.2-7.1 folds higher than the NF ratios of oxalate and malonate. However, this could be due to the formation of many other species as well as the matrix effects. We hesitate to speculate too much here. The SOA formation may have been underestimated due to the matrix
effect. SOA formation during aging of incense burning plume should be further assessed by other
quantitative online instruments in future works. We have added this clarification to the manuscript:

[Figure]

**Figure S15.** The NF ratios (aged to fresh) of oxalate and malonate, and TOC ratios as a function
of [$O_3$] in $O_3$+UV aging experiments.

Line 261-268: Compared to 300 ppb $O_3$ and UV, the NF ratios of malonate and oxalate were 30
and 9 folds higher, at 1500 ppb $O_3$ and UV, respectively. This trend is different from the
independence of nitrate formation on OH exposure, probably because the formation of SOA was
slower than nitrate via multiphase uptake. These NFs are lower estimates due to the possible
degradation by photolysis of Fe-decarboxylate complexes to $CO_2$ (Gen et al., 2021).While the
TOC ratio at different $O_3$+UV, which indicates the formation of SOA, shows an overall trend
similar to the NF ratios of oxalate and malonate (Figure S15), the TOC ratio was 1.2-7.1 folds
higher than the NF ratios of oxalate and malonate. However, this could be due to the formation of
many other species as well as the matrix effects. SOA formation during aging of incense burning
plume should be further assessed by other quantitative online instruments in future work.

*Technical Comments*

*Line 47: Awkward grammar in "For instance ... nitrate."*

**Author's Response:** We have revised the sentence as below:

Line 48-49: For instance, ozone and OH oxidations of NOx  were
considered primary sources of particulate nitrate (Seinfeld et al., 2008; Liang et al., 2021; Gen et
al., 2022).

*Line 325: Please check the citation styles; they are inconsistent.*

**Author's Response:** We have updated the citation styles according to the Endnote tool provided
by the website.

*Lines 409/412: Same reference cited twice?*

**Author's Response:** We have removed one of it.

*Figure 2: The data appears almost randomly scattered in the lower panels, which may be due to*
*points heavily overlapping with each other. The authors may want to replace the box and*
*whiskers/scatterplot with a violin plot for easier visualization.*

**Author's Response:** Thank you for the suggestions, we have replaced the box-whisker plot with
a violin plot.

[Figure]

**Figure 2.** (a) The whisker-box plot of total nitrate RPA of fresh and aged particles. The violin plots of (b, d) APA and (c, e) RPA of total nitrate in $O_3$- and OH-aged particles as a function of size (unit: μm) of particles aged at 1500 ppb $O_3$ (+ UV). The medians are shown as the lines, and the kernel densities represent the probability density of the data at different values.

*Figure S1: Please display where the charcoal absorber, HEPA filter, and NOx instrument would have been placed.*

**Author's Response:** The Figure S1 has been updated.

**References**

Cheng, C., Huang, Z., Chan, C. K., Chu, Y., Li, M., Zhang, T., Ou, Y., Chen, D., Cheng, P., and Li, L.: Characteristics and mixing state of amine-containing particles at a rural site in the Pearl River Delta, China, Atmospheric Chemistry and Physics, 18, 9147-9159, 2018.

Davis, D., Ravishankara, A., and Fischer, S.: SO2 oxidation via the hydroxyl radical: atmospheric fate of HSOx radicals, Geophysical Research Letters, 6, 113-116, 1979.

Denkenberger, K. A., Moffet, R. C., Holecek, J. C., Rebotier, T. P., and Prather, K. A.: Real-time, single-particle measurements of oligomers in aged ambient aerosol particles, Environmental Science & Technology, 41, 5439-5446, 2007.

Gen, M., Zhang, R., and Chan, C. K.: Nitrite/Nitrous Acid Generation from the Reaction of Nitrate and Fe(II) Promoted by Photolysis of Iron–Organic Complexes, Environmental Science & Technology, 55, 15715-15723, 10.1021/acs.est.1c05641, 2021.

Gen, M., Liang, Z., Zhang, R., Go Mabato, B. R., and Chan, C. K.: Particulate nitrate photolysis in the atmosphere, Environmental Science: Atmospheres, 10.1039/D1EA00087J, 2022.

Ji, X., Le Bihan, O., Ramalho, O., Mandin, C., D'Anna, B., Martinon, L., Nicolas, M., Bard, D., and Pairon, J.-C.: Characterization of particles emitted by incense burning in an experimental house, Indoor Air, 20, 147-158, https://doi.org/10.1111/j.1600-0668.2009.00634.x, 2010.

Kucinski, T. M., Dawson, J. N., and Freedman, M. A.: Size-Dependent Liquid–Liquid Phase Separation in Atmospherically Relevant Complex Systems, The Journal of Physical Chemistry Letters, 10, 6915-6920, 2019.

Le Breton, M., Psichoudaki, M., Hallquist, M., Watne, Å., Lutz, A., and Hallquist, Å.: Application of a FIGAERO ToF CIMS for on-line characterization of real-world fresh and aged particle emissions from buses, Aerosol Science and Technology, 53, 244-259, 2019.

Lee, S.-C. and Wang, B.: Characteristics of emissions of air pollutants from burning of incense in a large environmental chamber, Atmospheric Environment, 38, 941-951, https://doi.org/10.1016/j.atmosenv.2003.11.002, 2004.

Li, J., Liu, Q., Li, Y., Liu, T., Huang, D., Zheng, J., Zhu, W., Hu, M., Wu, Y., and Lou, S.: Characterization of aerosol aging potentials at suburban sites in northern and southern China utilizing a potential aerosol mass (Go: PAM) reactor and an aerosol mass spectrometer, Journal of Geophysical Research: Atmospheres, 124, 5629-5649, 2019.

Liang, Z., Zhang, R., Gen, M., Chu, Y., and Chan, C. K.: Nitrate Photolysis in Mixed Sucrose–Nitrate–Sulfate Particles at Different Relative Humidities, The Journal of Physical Chemistry A, 125, 3739-3747, 10.1021/acs.jpca.1c00669, 2021.

Liang, Z., Zhou, L., Infante Cuevas, R. A., Li, X., Cheng, C., Li, M., Tang, R., Zhang, R., Lee, P. K. H., Lai, A. C. K., and Chan, C. K.: Sulfate Formation in Incense Burning Particles: A Single-Particle Mass Spectrometric Study, Environmental Science & Technology Letters, 9, 718-725, 10.1021/acs.estlett.2c00492, 2022.

Liu, T., Zhou, L., Liu, Q., Lee, B. P., Yao, D., Lu, H., Lyu, X., Guo, H., and Chan, C. K.: Secondary Organic Aerosol Formation from Urban Roadside Air in Hong Kong, Environ Sci Technol, 53, 3001-3009, 10.1021/acs.est.8b06587, 2019.

Manoukian, A., Quivet, E., Temime-Roussel, B., Nicolas, M., Maupetit, F., and Wortham, H.: Emission characteristics of air pollutants from incense and candle burning in indoor atmospheres, Environmental Science and Pollution Research, 20, 4659-4670, 10.1007/s11356-012-1394-y, 2013.

Neubauer, K. R., Johnston, M. V., and Wexler, A. S.: Humidity effects on the mass spectra of single aerosol particles, Atmospheric Environment, 32, 2521-2529, 1998.

Passig, J., Schade, J., Irsig, R., Kröger-Badge, T., Czech, H., Adam, T., Fallgren, H., Moldanova, J., Sklorz, M., and Streibel, T.: Single-particle characterization of polycyclic aromatic hydrocarbons in background air in northern Europe, Atmospheric Chemistry and Physics, 22, 1495-1514, 2022.

Qin, X. and Prather, K. A.: Impact of biomass emissions on particle chemistry during the California Regional Particulate Air Quality Study, International Journal of Mass Spectrometry, 258, 142-150, https://doi.org/10.1016/j.ijms.2006.09.004, 2006.

Rebotier, T. P. and Prather, K. A.: Aerosol time-of-flight mass spectrometry data analysis: A benchmark of clustering algorithms, Analytica Chimica Acta, 585, 38-54, https://doi.org/10.1016/j.aca.2006.12.009, 2007.

Schilling, J. B.: Extraction of semivolatile organic compounds from high-efficiency particulate air (HEPA) filters by supercritical carbon dioxide, Argonne National Lab., Analytical Chemistry Lab., IL (United States), 1997.

Seinfeld, J. and Pandis, S.: Atmospheric chemistry and physics. 1997, New York, 2008.

Taiwo, A. M., Harrison, R. M., Beddows, D. C., and Shi, Z.: Source apportionment of single particles sampled at the industrially polluted town of Port Talbot, United Kingdom by ATOFMS, Atmospheric Environment, 97, 155-165, 2014.

Tsiligiannis, E., Hammes, J., Salvador, C. M., Mentel, T. F., and Hallquist, M.: Effect of NO x on 1, 3, 5-trimethylbenzene (TMB) oxidation product distribution and particle formation, Atmospheric chemistry and physics, 19, 15073-15086, 2019.

Wang, H., Guo, S., Wu, Z., Qiao, K., Tang, R., Yu, Y., Xu, W., Zhu, W., Zeng, L., and Huang, X.: Secondary organic aerosol formation from straw burning using an oxidation flow reactor, Journal of Environmental Sciences, 114, 249-258, 2022.

Watne, Å. K., Psichoudaki, M., Ljungström, E., Le Breton, M., Hallquist, M., Jerksjö, M., Fallgren, H., Jutterström, S., and Hallquist, Å. M.: Fresh and oxidized emissions from in-use transit buses running on diesel, biodiesel, and CNG, Environmental science & technology, 52, 7720-7728, 2018.

Xu, W., Li, Z., Lambe, A. T., Li, J., Liu, T., Du, A., Zhang, Z., Zhou, W., and Sun, Y.: Secondary organic aerosol formation and aging from ambient air in an oxidation flow reactor during wintertime in Beijing, China, Environmental Research, 209, 112751, 2022.

Yu, Y., Guo, S., Wang, H., Shen, R., Zhu, W., Tan, R., Song, K., Zhang, Z., Li, S., and Chen, Y.: Importance of semivolatile/intermediate-volatility organic compounds to secondary organic aerosol formation from Chinese domestic cooking emissions, Environmental Science & Technology Letters, 9, 507-512, 2022.

Zhang, Z., Zhu, W., Hu, M., Wang, H., Chen, Z., Shen, R., Yu, Y., Tan, R., and Guo, S.: Secondary organic aerosol from typical Chinese domestic cooking emissions, Environmental Science & Technology Letters, 8, 24-31, 2020.

Zhao, W., Hopke, P. K., and Prather, K. A.: Comparison of two cluster analysis methods using single particle mass spectra, Atmospheric Environment, 42, 881-892, https://doi.org/10.1016/j.atmosenv.2007.10.024, 2008.

Zhou, L., Liu, T., Yao, D., Guo, H., Cheng, C., and Chan, C. K.: Primary emissions and secondary production of organic aerosols from heated animal fats, Science of the Total Environment, 794, 148638, 2021a.

Zhou, L., Salvador, C. M., Priestley, M., Hallquist, M., Liu, Q., Chan, C. K., and Hallquist, Å. M.: Emissions and Secondary Formation of Air Pollutants from Modern Heavy-Duty Trucks in Real-World Traffic—Chemical Characteristics Using On-Line Mass Spectrometry, Environmental science & technology, 55, 14515-14525, 2021b.

---

## Author Comment (AC2)

**Response to Reviewer 2:**

*General Comments*

*This article describes a series of experiments used to explore the aging of incense particles in the presence of different oxidants by applying a flow reactor and a single particle mass spectrometer. The major conclusions are that nitrate formation on/in incense burning particles is enhanced in the presence of OH radicals relative to dark ozone oxidation and that oxalate and malonate formation is also enhanced in the presence of OH. While the experiments are interesting the main text and conclusions need more clarity around the actual implications of these findings for (presumably both indoor and outdoor) air quality and exposure.*

**Author's Response:** Thank you for the constructive comments. We have revised the manuscript accordingly and added clarifications to the main text and conclusions around the implications of the findings.

**Major comments**

*Section 3.2 is hard to follow and a clearer explanation and validation of the use of RPA and APA to draw conclusions is needed. A relatively low laser fluence was used in this work and partial ablation/matrix effects can be problematic for particles with secondary coatings (see Zelenyuk group research), which could lead to misclassification of particles based on ion intensities. The appearance of the nitrate-rich particle types after aging is quite conclusive though. Relative ionization/detection efficiencies for the different particle types are not discussed however, which would have a large impact on the relative particle type abundances shown in Figure 1. Some pure secondary organic particles (which may be externally mixed with primary particles in the reactor) for example are just not ionized effectively at all by SPMS. Some discussion of the drawbacks of SPMS are needed in the text.*

**Author's Response:** A control experiment doubling the laser fluence from 0.6 to 1.2 mJ showed minor differences in the classification of the aged particles and the RPA of total nitrate peaks (Table S1). This suggests that the partial ablation due to the formation of nitrate coating is not significant. It is possible that there were some pure secondary organic particles still not ionized at 1.2 mJ, which resulted in an underestimation of secondary organic aerosol formation. We have added this information to the manuscript.

**Table S1.** The classification of particles and RPA of total nitrate peaks of aged particles at different laser fluences.

| Experimental Conditions | Laser fluence (mJ) | OC-ON (%) | OC-N (%) | K-ON (%) | K-ONN (%) | K-N (%) | K-ONEC (%) | K-Cl (%) | $RPA_{Total\ nitrate}$ |
|---|---|---|---|---|---|---|---|---|---|
| O$_3$-dark-aged (800 ppb) | 0.6 | (7.6±1.0) | (21.9±1.7) | (29.0±0.7) | (11.2±0.9) | (2.5±0.5) | (22.8±1.8) | (3.9±0.4) | (0.40±0.15) |
| | 1.2 | (7.2±1.4) | (22.6±4.0) | (28.5±4.3) | (10.6±0.2) | (3.3±1.3) | (22.8±2.5) | (4.0±0.1) | (0.37±0.09) |
| O$_3$-UV-aged (800 ppb) | 0.6 | (0.0±0.0) | (36.7±7.2) | (0.0±0.0) | (26.7±2.1) | (35.5±4.2) | (0.0±0.0) | (0.0±0.0) | (0.66±0.03) |
| | 1.2 | (0.0±0.0) | (35.6±2.3) | (0.0±1.0) | (28.6±1.9) | (33.6±4.7) | (0.2±0.2) | (0.0±0.0) | (0.67±0.07) |

Line 229-232: A control experiment doubling the laser fluence from 0.6 to 1.2 mJ showed minor differences in the classification of the aged particles and the RPA of total nitrate peaks (Table S1). This suggested that the partial ablation due to the formation of nitrate coating is not significant. It is possible that there were some pure secondary organic particles still not ionized at 1.2 mJ, which resulted in an underestimation of secondary organic aerosol formation.

*Where is the NOx added? Or is it just the NOx from the combustion process. How were the NOx concentrations monitored or validated to be similar between experiments?*

**Author's Response:** The NOx was from the combustion process. We have added that the location of NOx monitor to the schematic figure (Figure S1).

[Figure]

**Figure S1.** The schematic of the experimental set-up. The NOx analyzer and the VOC sensor were used only in the experiment for determining the gas removal efficiency and NOx decay under OH exposure.

*Figure S1 is missing the scrubber mentioned in the text.*

**Author's Response:** We have added the scrubber location to Figure S1.

*Why are most of the figures in the supplementary information? There are only 3 main text figures and a lot of reference to the supplementary figures.*

**Author's Response:** The main text figures are combined ones, converting the key messages of the manuscript. The supplementary figures play minor roles. The key results are shown in the manuscript.

*The manuscript is hard to follow in places, including the conclusions. A clear central message needs to be stated in the conclusions, with implications for air quality or public health.*

**Author's Response:** We have added a central message and implications.

Line 292-295: In this work, we report the single-particle mixing state characteristics of incense burning particles upon ozonolysis and photochemical oxidation. Formation of secondary aerosol including nitrate and organics was found. This indicates that besides the significant primary emission of particles, additional particulate pollutants could be formed upon atmospheric aging, further worsening the air quality in both outdoor and indoor environments.

*No correlation (mentioned in the caption) is shown in Figure S4. Same for S6*

**Author's Response:** We have corrected the captions.

**Figure S7.** APA of ON as a function of APA of total nitrate in fresh and $O_3$-aged particles.

**Figure S9.** RPA of total nitrate as a function of RPA of formate.

*Why was no support analysis done (especially ion chromatography for the oxalate/malonate yields) to rule out potential matrix-effect artifacts. Quantification (even semi-quantification) exclusively using SPMS is associated with high uncertainty.*

**Author's Response:** We have added TOC analysis to the water-extract of collected particles. While the TOC ratio at different $O_3$+UV shows an overall trend similar to the NF ratios of oxalate and malonate, which indicates the formation of SOA, the TOC ratio was 1.2-7.1 folds higher than the NF ratios of oxalate and malonate (Figure S15). However, this could be due to the formation of many other species as well as the matrix effects. We hesitate to speculate too much here. SOA formation during aging of incense burning plume should be further assessed by other quantitative online instruments in future works. We have added this clarification to the manuscript:

[Figure]

**Figure S15.** The NF ratios (aged to fresh) of oxalate and malonate, and TOC ratios as a function of [O₃] in O₃+UV aging experiments.

Line 264-268: While the TOC ratio at different O₃+UV shows an overall trend similar to the NF ratios of oxalate and malonate, which indicates the formation of SOA, the TOC ratio was 1.2-7.1 folds higher than the NF ratios of oxalate and malonate (Figure S15). However, this could be due to the formation of many other species as well as the matrix effects. SOA formation during aging of incense burning plume should be further assessed by other quantitative online instruments in future work.

***Specific comments***

*Abstract- remove (photo-) because dark oxidation is also investigated*

**Author's Response:** The (photo-) has been removed.

*Page 2 line 58, missing 'tracers'*

**Author's Response:** We have removed the empty row line 58.

*"We studied the aging of the particles under 'UV', 'O3 and dark', and 'O3 and UV' in the PAM. Since UV at 254 nm is expected 81 to photolyze O3 to form OH radicals in the presence of water vapor, we named these aged particles UV-aged, O3-aged, and 82 OH-aged, respectively." Why not: 'UV-aged', 'O₃-dark-aged' 'O₃-UV-aged' for clarity*

**Author's Response:** We have replaced all 'O₃-aged' and 'OH-aged' with 'O₃-dark-aged' and 'O₃-UV-aged'.

*Where are the nitrate peaks in Figure 1b?*

**Author's Response:** Figure 1b showed the 'organic difference spectra' with EC and inorganic removed. We have further clarified the contents of the figure.

Line 152-157: To compare the changes in the organic signals, we first excluded all inorganics and EC peaks (Table S2). Control experiments atomizing $KNO_3$ solution (as $K^+$ is the main inorganic cation found in incense burning particles) showed the RPA ratio of -16[O] to nitrate peaks is (6 ± 1.7) % due to fragmentation. Sulfate shows negligible fragmentation under our experimental conditions (Liang et al., 2022). Thus, we subtracted the RPA of -16[O] by 6% RPA of nitrate. Then, we recalculated the relative peak area (RPA) of the organic peaks only, defined as "organic spectra". Figure 1b shows the differences in the organic spectra of the aged and fresh particles.

*"The instrument 98 was routinely calibrated with polystyrene latex spheres of 0.2-2.5 μm diameter (Nanosphere Size Standards, Duke Scientific 99 Corp., Palo Alto)" It should be specified that this only calibrates the sizing accuracy, not chemical species. "and more than 98% of the particles were analyzed" should be "were classified"*

**Author's Response:** We have added clarification and revised the text:

Line 110-112: The size accuracy of the instrument was routinely calibrated with polystyrene latex spheres of 0.2-2.5 μm diameter (Nanosphere Size Standards, Duke Scientific Corp., Palo Alto).

Line 114: more than 98% of the particles were classified.

*""K-" particles contain a dominant +39 peak and a small +41 peak attributed to isotopic 116 potassium (Bi et al., 2011). On the other hand, the "OC-" particles are rich in typical organic fragments such as +27[C2H3] (Cheng et al., 2017). According to the negative spectra, "-ON" particles have dominant ON signals. "-ONEC" particles have elemental carbon (EC) peaks of -12n[Cn - ], with intensities comparable to typical ON peaks (Zhou et al., 2020). "-Cl" particles have prominent Cl- (m/z=-35, -37(isotopic)) and KCl2 - (m/z=-109, -111(isotopic)) peaks (Bi et al., 2011)." These are not the correct references for the original identification of these peaks by single particle mass spectrometry. See Prather group research.*

**Author's Response:** We have replaced the reference with those for original identification.

Line 135-139: ""K-" particles contain a dominant +39 peak and a small +41 peak attributed to isotopic potassium (Silva et al., 1999). On the other hand, the "OC-" particles are rich in typical organic fragments such as +27[$C_2H_3$] (Silva et al., 2000). According to the negative spectra, "-ON" particles have dominant ON signals. "-ONEC" particles have elemental carbon (EC) peaks of -12n[$C_n^-$], with intensities comparable to typical ON peaks (Whiteaker et al., 2002). "-Cl" particles have prominent $Cl^-$ (m/z=-35, -37(isotopic)) and $KCl_2^-$ (m/z=-109, -111(isotopic)) peaks (Guazzotti et al., 2001; Dall'osto et al., 2004)."

"except for the rise of -62[NO3 - ] and -46[NO2 -], which indicates the formation of nitrate and probably nitrite" Note that -46 is observed from nitrate and can't be used to confirm nitrite

**Author's Response:** We agreed that $-62[NO_3^-]$ can be fragmented into $-46[NO_2^-]$, but we can not exclude the existence of nitrite. Therefore, we used 'probably' for conservation.

*"However, the RPA shows an opposite trend, which can be interpreted as lower nitrate concentration in larger particles. Larger particles have larger surfaces but smaller surface-to-volume ratios, which lead to a larger absolute amount of nitrate formed but a lower relative concentration of particulate nitrate (Figure 2c, e). Under O3+UV, it is also possible that comparable HNO3 was generated under excess [OH] and contributed to the similar total nitrate RPA since the [NOx] reductions under different OH exposure are similarly high (Figure S10). The insensitivity of nitrate formation to O3 and OH exposure can be potentially explained by the diffusion limitation of interfacial uptake due to the poor hygroscopicity of fresh incense burning particles (Li and Hopke, 1993; Zaveri et al., 2018; Slade and Knopf, 2014; 220 Liang et al., 2022a)"*

*Caution should be taken interpreting the RPAs due to the potential for partial ablation and matrix effects.*

**Author's Response:** Control experiments using different laser fluence indicate that partial ablation is insignificant. We have further clarified that the RPA is not equivalent to concentration due to the matrix effect.

Line 119-122: The relative peak area (RPA), defined as the peak area of a specific peak divided by the total positive or negative mass spectral peak area, can reflect the relative abundance of particulate components (Liang et al., 2022). Note that RPA is not equivalent to concentration due to the matrix effect. Furthermore, control experiments using different laser fluences indicate that partial ablation was insignificant (Table S1).

*"Figure 3a shows the NF ratio (aged particles to fresh particles) of oxalate and malonate. We used the NF ratio rather than the APA or RPA, to avoid large uncertainties in organic abundance due to the much weaker peaks of organics in the spectra." For the number fraction is there a query using a minimum APA or RPA for oxalate or malonate to be classified as a count? Furthermore, the point made here does raise the issue with using APAs/RPAs at all considering the potential for matrix effects.*

**Author's Response:** The APA threshold for oxalate or malonate was 15 arbitrary unit (a.u.) with a baseline of zero unit, the same as that used in ambient measurements (Zhu et al., 2020). We have clarified this and mentioned the potential matrix effects.

Line 256-259: We used the NF ratio rather than the APA or RPA to avoid large uncertainties in organic abundance due to the much weaker peaks of organics in the spectra, which was potentially

due to the matrix effects. The APA threshold for oxalate or malonate was 15 arbitrary unit (a.u.) with a baseline of zero unit, the same as that adopted in an ambient study (Zhu et al., 2020).

**References**

Dall'Osto, M., Beddows, D. C. S., Kinnersley, R. P., Harrison, R. M., Donovan, R. J., and Heal, M. R.: Characterization of individual airborne particles by using aerosol time-of-flight mass spectrometry at Mace Head, Ireland, Journal of Geophysical Research: Atmospheres, 109, https://doi.org/10.1029/2004JD004747, 2004.

Guazzotti, S. A., Coffee, K. R., and Prather, K. A.: Continuous measurements of size-resolved particle chemistry during INDOEX-Intensive Field Phase 99, Journal of Geophysical Research: Atmospheres, 106, 28607-28627, https://doi.org/10.1029/2001JD900099, 2001.

Liang, Z., Zhou, L., Infante Cuevas, R. A., Li, X., Cheng, C., Li, M., Tang, R., Zhang, R., Lee, P. K. H., Lai, A. C. K., and Chan, C. K.: Sulfate Formation in Incense Burning Particles: A Single-Particle Mass Spectrometric Study, Environmental Science & Technology Letters, 9, 718-725, 10.1021/acs.estlett.2c00492, 2022.

Silva, P. J. and Prather, K. A.: Interpretation of Mass Spectra from Organic Compounds in Aerosol Time-of-Flight Mass Spectrometry, Analytical Chemistry, 72, 3553-3562, 10.1021/ac9910132, 2000.

Silva, P. J., Liu, D.-Y., Noble, C. A., and Prather, K. A.: Size and Chemical Characterization of Individual Particles Resulting from Biomass Burning of Local Southern California Species, Environmental Science & Technology, 33, 3068-3076, 10.1021/es980544p, 1999.

Whiteaker, J. R., Suess, D. T., and Prather, K. A.: Effects of Meteorological Conditions on Aerosol Composition and Mixing State in Bakersfield, CA, Environmental Science & Technology, 36, 2345-2353, 10.1021/es011381z, 2002.

Zhu, S., Li, L., Wang, S., Li, M., Liu, Y., Lu, X., Chen, H., Wang, L., Chen, J., Zhou, Z., Yang, X., and Wang, X.: Development of an automatic linear calibration method for high-resolution single-particle mass spectrometry: improved chemical species identification for atmospheric aerosols, Atmos. Meas. Tech., 13, 4111-4121, 10.5194/amt-13-4111-2020, 2020.

---

## Author Comment (AC3)

**Secondary Aerosol Formation in Incense Burning Particles by Ozonolysis and Photochemical Oxidation via Single Particle Mixing State Analysis**

Zhancong Liang[1,2], Liyuan Zhou[1,2], Xinyue Li[1], Rosemarie Ann Infante Cuevas[1,2], Rongzhi Tang[1,2], Mei Li[3,4], Chunlei Cheng[3,4], Yangxi Chu[5], Patrick. K.H. Lee[1], Alvin. C.K. Lai[1], Chak K. Chan[1,2,6]*

[revised manuscript text omitted]

**2 Experimental**

**2.1 Aging of incense-burning particles**

The schematic of the experimental set-up can be found in Figure S1. In brief, we burnt an incense stick (Figure S2, Kwok Tin

Heung, Hong Kong) in a 20 L glass burning bottle for each experiment. The air exchange rate per hour (ACH) is 0.3, comparable to the typical natural ventilation conditions (Lee et al., 2004). The relative humidity (RH) and the temperature inside the burning bottle were 56 ± 9 % RH and 22 ± 2.7 °C. The burning was rapidly converted from flaming to smoldering after ignition. A two-stage system diluted the emissions with an overall dilution of around 1600. Compressed air (~0.1 L min$^{-1}$) was used to introduce the diluted incense burning particles to the PAM reactor equipped with two UVC light tubes (30W, Philips TUV, $\lambda$max = 254nm). The spectrum of the lamp was shown in Supplementary information (Figure S3). In the control experiments, a charcoal absorber or HEPA filter was used to remove the gaseous pollutants or particles prior to the introduction to the Go:PAM. The data of these control experiments can be found in the supplementary information (Figure S4, S5). All these control experiments were aged experiments. The removal efficiency of NOx, VOCs, and particles were ~85%, ~90%, and ~100%, respectively, determined by the concentration reduction after applying a HEPA filter or charcoal absorber at the exhaust of the burning bottle, using a NOx analyzer (T200, Teledyne) or a Total VOC analyzer (Yuante) (Figure S1). A controlled dry-wet mixed carrier flow of compressed air (~4 L min$^{-1}$) and a flow of $O_3$ (~0.1 L min$^{-1}$) generated by passing $O_2$ (99.995%, Linde) to an $O_3$ generator (Model 610, Jelight Company Inc, USA) were introduced into the Go:PAM. The compressed air was treated by a HEPA filter and a charcoal absorber prior to the experiment system. [$O_3$] ranged from 300 to 1500 ppb, equivalent to an atmospheric ozone exposure of 10-50 min, assuming ambient concentration of 60 ppb (Xia et al., 2021). The RH at the exit of the PAM was monitored by an RH sensor (M170, Vaisala, Finland). All the experiments were conducted at 80% RH and 22 $\pm$ 1.7 °C. The residence time in the Go:PAM was ~100 s. The exhaust of the PAM was characterized by an $O_3$ analyzer (106L, 2B technology, USA), a water-based condensation particle counter (WCPC, Aerosol dynamics Inc, USA), and a SPAMS (Hexin Analytical Instrument Co., Ltd, China). Watne et al. suggested that the penetration of the particles is close to 100% for particles larger than 100 nm. Hence the wall loss is negligible for the 0.2-2 um particles that SPAMS measures. Besides, a control experiment measuring the total VOCs at the entrance and the exhaust of the Go:PAM suggested that the gas wall loss was also minor (6 $\pm$ 4%). The particles passed through a diffusion dryer before entering the Go:PAM to reduce the matrix effects from water (Neubauer et al., 1998). The RH at the exhaust of the diffusion dryer was ~15%. The residence time of the particles in the dryer was estimated to be 5 s and the particle loss was ~4% according to the CPC measurements. We also collected particles on 47 mm quartz filters (PALL, USA) at the exhaust of the Go:PAM reactor for offline mass and chemical analysis. The number of particles collected on the filters was estimated by the total WCPC counts during the sampling period. The particle number concentration from the WCPC was 6100 $\pm$ 2400 # cm$^{-3}$ and the estimated number of collected particles was around $10^8$ #. The filter sample was extracted by deionized (DI) water for analyzing water-soluble ions (e.g., nitrate, formate, potassium) by Ion chromatography (IC) using the same protocol reported in our previous work (Liang et al., 2022). The total organic content (TOC) of the water-extract was analyzed by a TOC analyzer (Shimadzu TOC-L).

We studied the aging of the particles under 'UV', '$O_3$ and dark', and '$O_3$ and UV' in the PAM. We named these aged particles UV-aged, $O_3$-Dark-aged, and $O_3$-UV-aged, respectively. Although 254 nm is not atmospherically relevant, UV-aged particles are used as a reference in the discussions of the properties of $O_3$-UV-aged particles. The OH exposure, equivalent to the product of gas-phase OH concentration and residence time, was determined by introducing a stream of $SO_2$ to the PAM for consuming OH radicals and monitoring the [$SO_2$] decay, following a well-established approach in the literature (Kang et al., 2007). [$SO_2$]

was almost constant under UV on but without $O_3$, suggesting that the photochemistry of incense plume does not affect our estimation of OH exposure. The upper limit of OH exposure used in this study varied from $1 \times 10^{10}$ to $5 \times 10^{10}$ molecules cm$^{-3}$ s, equivalent to 2~10 hours of photochemical aging, assuming an ambient OH concentration of $1.5 \times 10^6$ molecules cm$^{-3}$

(Mao et al., 2009).

**2.2 SPAMS analysis**

A detailed description of the SPAMS can be found in Li et al (Li et al., 2011). After the particle flow exits the PAM reactor, it first passes a PM$_{2.5}$ cyclone to avoid clogging before entering the SPAMS through a 0.1 mm critical orifice at 80 mL min$^{-1}$

flow. Particles achieved a terminal velocity in the supersonic expansion airflow and were detected and aerodynamically sized by two continuous diode Nd: YAG laser beams (532 nm). They were then ionized by a pulsed Nd: YAG laser (266 nm)

triggered based on the velocity of a specific particle. The positive and negative ions produced were detected according to the different mass-to-charge ratios (m/z). The energy of the ionization laser was kept at ~0.6 mJ (Cheng et al., 2017). Spectra of more than 3000 individual particles collected for ~15 min were used for further analysis for each experiment. The size accuracy of the instrument was routinely calibrated with polystyrene latex spheres of 0.2-2.5 μm diameter (Nanosphere Size Standards,

Duke Scientific Corp., Palo Alto). An adaptive resonance theory method (ART-2a) based on MATLAB was used to categorize the incense particles of similar SPAMS spectral characteristics into different particle groups (Phares et al., 2001). In the ART-

2a analysis, we used a vigilance factor of 0.85, and more than 98% of the particles were classified. Note that there is no general rule for the vigilance factor in ART-2a. Zhao et al. (2008) reported that both a small vigilance factor (e.g., 0.5 or 0.6) and a relatively high vigilance factor (e.g., 0.8) show very similar clustering accuracies (± ~5%) (Zhao et al., 2008).

**3 Results and discussions**

**3.1 Single-particle characteristics of incense burning particles**

The relative peak area (RPA), defined as the peak area of a specific peak divided by the total positive or negative mass spectral peak area, can reflect the relative abundance of particulate components (Liang et al., 2022). Note that RPA is not equivalent to concentration due to the matrix effect.  Furthermore, control experiments using different laser fluences indicate that partial ablation was insignificant (Table S1, will be discussed later). The average spectra of the incense burning particles (Figure 1a)

are similar to our previous work on incense burning at 50% RH (Liang et al., 2022). +39[K] dominates the positive spectra, and organic nitrogen (ON) peaks (i.e., -26[CN] and -42[CNO]) from nitrogen-containing organics (NOC) dominate the negative spectra (Zhang et al., 2020; Zhai et al., 2015; Zhang et al., 2021). These features are also found in biomass burning particles (Bi et al., 2011; Peng et al., 2019; Luo et al., 2020). Despite that particulate polyaromatic hydrocarbons (PAHs) were found in a previous incense combustion study (Ji et al., 2010)  and a recent study of ambient particles based on a single-particle mass spectrometer with ART-2a (Passig et al., 2022), none of the fresh incense burning particles in our experiments contained the PAHs peaks (m/z = 178, 189 (fragment of alkylated phenanthrenes), 202, 220, 228, and 252). Regardless of the presence of PAHs or not, our conclusion on nitrate formation does not depend on the detection of specific chemicals such as PAHs.

ART-2a categorizes fresh incense burning particles into K-ON, K-ONEC, K-Cl, and OC-ON. EC, Cl and OC are abbreviations of elemental carbon, chloride and organic carbon, respectively. Briefly, the "K" and "OC" before the hyphen indicate the characteristics of the positive spectra, while "ON", "ONEC" and "Cl" after the hyphen indicate the characteristics of the negative spectra. "K-" particles contain a dominant +39 peak and a small +41 peak attributed to isotopic potassium (Silva et al., 1999). On the other hand, the "OC-" particles are rich in typical organic fragments such as $+27[C_2H_3]$ (Silva et al., 2000).

According to the negative spectra, "-ON" particles have dominant ON signals. "-ONEC" particles have elemental carbon (EC)

peaks of $-12n[C_n^-]$, with intensities comparable to typical ON peaks (Whiteaker et al., 2002). "-Cl" particles have prominent

$Cl^-$ (m/z=-35, -37(isotopic)) and $KCl_2^-$ (m/z=-109, -111(isotopic)) peaks (Guazzotti et al., 2001; Dall'osto et al., 2004). The average spectra of each category can be found in Figure S6. There are slightly fewer K-ON particles and more K-ONEC

particles observed at 80% RH (this work) than at 50% RH in Liang et al (Liang et al., 2022), probably due to the lower organic concentrations at higher RH to limit particle-phase partitioning of volatile organic compounds (Donaldson et al., 2006; Mcfall et al., 2020; Chan et al., 2011; Chan et al., 2010). Overall, the number fraction (NF) of each category is similar to our previous work, with a descending order of K-ON (47.3±5.2%) >> OC-ON (25.7±4.7%) ≈ K-ONEC (20.2±2.8%) > K-Cl (5.1±1.1%)

(Figure 1c), reflecting the fresh incense burning particles are organic-rich (Li et al., 2012; Zhang et al., 2022).

**3.2 Ozonolysis of the incense burning particles**

Figure 1a also shows the average spectra of aged incense burning particles under 800 ppb $O_3$. Qualitatively, the major peaks are similar to those in fresh incense burning particles, except for the rise of $-62[NO_3^-]$ and $-46[NO_2^-]$, which indicates the formation of nitrate and probably nitrite. The formation of organo-nitrate is not considered significant due to the decreased -

26[CN] and -42[CNO].

To compare the changes in the organic signals, we first excluded all inorganics and EC peaks (Table S2). Control experiments atomizing $KNO_3$ solution (as $K^+$ is the main inorganic cation found in incense burning particles) showed the RPA ratio of -

16[O] to nitrate peaks is (6 ± 1.7) % due to fragmentation. Sulfate shows negligible fragmentation under our experimental conditions (Liang et al., 2022). Thus, we subtracted the RPA of -16[O] by 6% RPA of nitrate. Then, we recalculated the RPA

of the organic peaks only, defined as "organic spectra". Figure 1b shows the differences in the organic spectra of the aged and fresh particles. The positive difference spectra show an RPA increase in the hydrocarbon $+37[C_3H]$ but an RPA decrease in

[revised manuscript text omitted]

**3.3 Photochemical oxidation of incense burning particles**

With UV (254nm) on, the 800 ppb $O_3$ was partly photolyzed to generate OH radicals in the presence of water vapor, resulting in an OH exposure of ~3× $10^{10}$ molecules cm$^{-3}$ s, equivalent to a photochemical age of ~5 h. We will use xx ppb $O_3$ (initial concentration) +UV, instead of OH exposure, to describe OH aging. The average spectra of $O_3$-UV-aged particles are generally similar to that of $O_3$-Dark-aged particles, with potassium and nitrate peaks dominating the positive and negative spectra, respectively (Figure 1a). However, the RPA of -46[$NO_2$] and -62[$NO_3$] were 0.2 and 0.4, around 2 times higher than $O_3$-Dark-aged particles, likely indicating more nitrate formation. As will be discussed later, photochemistry triggered by light-absorbing compounds such as photosensitizers and Fe salts is a possible source of nitrate formation in $O_3$-UV-aged particles.[51-53] However, its contribution is considered minor compared with OH chemistry since UV-aged particles only show a total nitrate RPA of 0.05, much lower than that of $O_3$-UV-aged particles (~ 0.7, will be discussed later). Control experiments using a charcoal absorber to remove the NOx significantly reduced the RPA of total nitrate by ~75% (Figure S4). These suggest that OH chemistry involving NOx dominated the particulate nitrate formation under OH exposure. Under 800 ppb $O_3$ and UV, the ~90% reduction of [NOx] with a simultaneous increase in total nitrate peaks under UV suggests the oxidation of NOx by OH radicals to form $HNO_2$ and $HNO_3$, which can be uptake by the particles afterward (Finlayson-Pitts et al., 1999). Reactive uptake of NOx initiated by OH chemistry cannot be excluded.

Similar to the $O_3$-Dark-aged particles, $O_3$-UV-aged particles show decreases in ON and other organic peaks (+38[$C_3H_2$], +50[$C_4H_2$], and +51[$C_4H_3$]) in the difference organic averaged spectra (Silva et al., 2000), likely due to oxidative consumption by OH radicals (Figure 1b). The ON peaks decrease in $O_3$-UV-aged particles was more significant than in $O_3$-Dark-aged particles, whereas the increase in formate peak is less obvious. These indicate that NOCs can also be effectively degraded via OH oxidation. Using the commonly used general markers of oxidized/aged organics in single-particle mass spectrometric studies of -16[O], -17[OH], +42[$C_2H_2O$], and +43[$C_2H_3O$] as examples (Taiwo et al., 2014; Denkenberger et al., 2007; Qin et al., 2006), the RPA increase in $O_3$-UV-aged particles are 18, 10, 3, and 17 times higher than in $O_3$-Dark-aged particles. This suggests that OH aging produced more oxidized products than $O_3$ aging. The difference average organic spectra of UV-aged particles almost showed no noticeable peaks, indicating that the chemistry initiated by particulate photoactive compounds may not be essential to the transformation of the organics (Figure S12).

The $O_3$-UV-aged particles can be categorized into K-ONN, K-N, and OC-N, and they generally have more intense nitrate peaks than $O_3$-Dark-aged particles. Still, "-ONN" particles have comparable ON and nitrate peaks, and "-N" particles have dominant nitrate peaks in the negative spectra (Figure S6). The NF descends in the order of OC-N (35.7±7.2%) ≈ K-N (35.5±4.2%) > K-ONN (25.7±2.1%) (Figure 1c). Notably, the NF of OC- particles of $O_3$-UV-aged particles is 50% larger than the fresh particles, likely due to the formation of additional particulate organics. We could not identify any preferential nitrate formation in specific particle types since most of the particles have high RPA of nitrate. The collection efficiency of SPAMS

increased from ~1% at 200 nm to ~40% at 960 nm (Figure S13). Therefore, secondary aerosol formation in small particles may have been underestimated. However, this would not affect the conclusion of the results that nitrate formed in incense burning particles upon $O_3$+Dark and $O_3$+UV aging. A control experiment doubling the laser fluence from 0.6 to 1.2 mJ showed minor differences in the classification of the aged particles and the RPA of total nitrate peaks (Table S1). This suggested that the partial ablation due to the formation of nitrate coating is not significant. It is possible that there were some pure secondary organic particles still not ionized at 1.2 mJ, which resulted in an underestimation of secondary organic aerosol formation.

**3.4 The formation of secondary nitrate**

Figure 2a shows the RPA of nitrate peaks under UV and different exposure of $O_3$ and OH. Since fresh particles also have high

NF of total nitrate, NF cannot accurately depict the effectiveness of nitrate formation. Fresh incense burning particles exhibit very low RPA of total nitrate, whereas exposure to $O_3$ increases the RPA from almost 0 to around 0.2, irrespective of the $[O_3]$.

Only a slight increase (~0.02) in total nitrate RPA was observed for UV-aged particles. However, with both $O_3$ and UV on, the RPAs of total nitrate further increased to above 0.7, which is also independent of the initial $[O_3]$. Consistent with the average spectra shown before, nitrate formation due to OH oxidation is likely more efficient than that by ozonolysis. Under both $O_3$ and OH exposure, the summed APA of nitrate peaks increased as particle size increased, suggesting possibly a larger total amount of nitrate formed in larger particles (Figure 2b, d). However, the RPA shows an opposite trend, which can be interpreted as lower nitrate concentration in larger particles. Larger particles have larger surfaces but smaller surface-to-volume ratios, which lead to a larger absolute amount of nitrate formed but a lower relative concentration of particulate nitrate (Figure

2c, e). It has been reported that the larger organic and inorganic mixed particles could undergo liquid-liquid phase separation (LLPS) more easily than smaller ones (Kucinski et al., 2019). It is possible that the organics are mainly located at the outer layers of the particles, whereas the inorganic components reside in the core. The hydrophobic organic shell may have retarded the uptake of $HNO_3$/$HNO_2$/$NOx$ to form nitrate. Under $O_3$+UV, it is also possible that comparable $HNO_3$ was generated under excess [OH] and contributed to the similar total nitrate RPA since the [NOx] reductions under different OH exposure are similarly high (Figure S14). The insensitivity of nitrate formation to $O_3$ and OH exposure can be potentially explained by the diffusion limitation of interfacial uptake due to the poor hygroscopicity of fresh incense burning particles (Li et al., 1993;

Zaveri et al., 2018; Slade et al., 2014; Liang et al., 2022).

**3.5 The Potential formation of SOA**

Oxalate and malonate are two major dicarboxylates in atmospheric particles and are considered SOA (Yao et al., 2002). They have been widely studied using single-particle mass spectrometry with well-validated detection efficiency, without severe complications in mass spectra due to fragmentations (Cheng et al., 2017; Sullivan et al., 2007). Figure 3a shows the NF ratio (aged particles to fresh particles) of oxalate and malonate. We used the NF ratio rather than the APA or RPA to avoid large uncertainties in organic abundance due to the much weaker peaks of organics in the spectra, which was potentially due to the matrix effects. The APA threshold for oxalate or malonate was 15 arbitrary unit (a.u.) with a baseline of zero unit, the same as that adopted in an ambient study (Zhu et al., 2020).

Compared to 300 ppb $O_3$ and UV, the NF ratios of malonate and oxalate were 30 and 9 folds higher, at 1500 ppb $O_3$ and UV, respectively. This trend is different from the independence of nitrate formation on OH exposure, probably because the formation of SOA was slower than nitrate via multiphase uptake. These NFs are lower estimates due to the possible degradation by photolysis of Fe-decarboxylate complexes to $CO_2$ (Gen et al., 2021). While the TOC ratio at different $O_3$+UV which indicates the formation of SOA, shows an overall trend similar to the NF ratios of oxalate and malonate (Figure S15), the TOC

ratio was 1.2-7.1 folds higher than the NF ratios of oxalate and malonate. However, this could be due to the formation of many other species as well as the matrix effects. SOA formation during aging of incense burning plume should be further assessed by other quantitative online instruments in future work. In contrast, no oxalate and malonate were observed during ozonolysis, irrespective of [$O_3$]. It is likely the SOA characteristics and precursors are different under $O_3$ and OH. Hence, we are conservative and use the term "potential", even though oxalate and malonate are commonly found highly oxidized SOA. The

TOC ratio of aged to fresh particles extract was higher upon OH oxidation than $O_3$ oxidation (Figure S16). Furthermore, UV- aged particles did not show an NF increase of both, indicating that the oxalate and malonate formation were mainly due to OH

[revised manuscript text omitted]

---

## Author Comment (AC4)

Supplementary Information to

**Secondary Aerosol Formation in Incense Burning Particles by Ozonolysis and Photochemical Oxidation via Single Particle Mixing State Analysis**

Zhancong Liang[1,2], Liyuan Zhou[1,2], Xinyue Li[1], Rosemarie Ann Infante Cuevas[1,2], Rongzhi Tang[1,2], Mei Li[3,4], Chunlei Cheng[3,4], Yangxi Chu[5], Patrick. K.H. Lee[1], Alvin. C.K. Lai[1], Chak K. Chan[1,2,6]*

[1] School of Energy and Environment, City University of Hong Kong, Hong Kong, China

[2] City University of Hong Kong Shenzhen Research Institute, Shenzhen, China

[3] Institute of Mass Spectrometry and Atmospheric Environment, Guangdong Provincial Engineering Research Center for On-line Source Apportionment System of Air Pollution, Jinan University, Guangzhou 510632, China

[4] Guangdong-Hongkong-Macau Joint Laboratory of Collaborative Innovation for Environmental Quality, Guangzhou 510632, China

[5] State Key Laboratory of Environmental Criteria and Risk Assessment, Chinese Research Academy of Environmental Sciences, Beijing, 100012, China

[6] Low-Carbon and Climate Impact Research Centre, City University of Hong Kong, Hong Kong, China

*To whom the correspondence should be addressed: *chak.k.chan@cityu.edu.hk*

**Content**

**Text S1.** Estimation of the OH exposure.

**Text S2.** Estimation of the mass hygroscopic growth factor (GF) by AIOMFAC model.

**Text S3.** The evolution of organic fragments.

**Figure S1.** Schematic of the experimental set-up.

**Figure S2.** The appearance of the incense sticks.

**Figure S3.** The emission spectrum of the UVC lamps.

**Figure S4.** RPA of summed nitrate peaks of aged incense burning particles in the presence and absence of charcoal absorber.

**Figure S5.** NF of oxalate and malonate of $O_3$-UV-aged incense burning particles in the presence and absence of a charcoal absorber.

**Figure S6.** Positive and negative spectra of different categorizations of particles.

**Figure S7.** APA of ON as a function of APA of total nitrate in fresh and $O_3$-Dark-aged particles.

**Figure S8.** $[NO_2^-]/[NO_3^-]$ in water-extract of aged particles.

**Figure S9.** RPA of total nitrate as a function of RPA of formate.

**Figure S10.** $[Formate]/[K^+]$ as a function of $[NO_3^-]/[K^+]$ in the water-extract of fresh and $O_3$-Dark-aged incense burning particles.

**Figure S11**. The whisker-box plot of the $[K^+]$ in the particle water-extract measured by IC normalized by the total counts of collected particles for IC measurement.

**Figure S12.** The difference of the average organic spectra of UV-aged particles and the NF ratio of UV-aged particles to fresh particles.

**Figure S13.** Relation between the particle collection efficiency of Polystyrene Latex particles and diameter in SPAMS.

**Figure S14.** Evolution of [NOx] under different OH exposure as a function of time.

**Figure S15.** The NF ratios of oxalate and malonate, and TOC ratios as a function of $[O_3]$ in $O_3$+UV aging experiments.

**Figure S16.** The total organic content (TOC) ratios of aged to fresh particles extract.

**Figure S17.** Size distribution of $O_3$-UV-aged particles and their dicarboxylate-containing fractions.

**Figure S18.** The GF of KCl and $KNO_3$ particles as a function of RH.

**Figure S19.** The NF ratio of aged particles to fresh particles.

**Figure S20.** The NF of new organic peaks in aged particles.

**Table S1.** The classification of particles and RPA of total nitrate peaks of aged particles at different laser fluences.

**Table S2.** Potential peaks from the inorganics and elemental carbons.

**Text S1.** Estimation of the OH exposure.

$SO_2$ was used to calculate the OH exposure in the Go:PAM. The UVC lamps were turned on to warm up for ~30 min and turned off. Then, $O_3$ (300, 800, 1500 ppb) and $SO_2$ (~200 ppb) were introduced to the Go:PAM with the UVC lamps turned off until its initial concentration remained constant at steady-state conditions, which typically took around 5 min. The $[SO_2]$ was recorded as $[SO_2]_{Initial}$. After that, the UVC lamps were turned on until the final $[SO_2]$ stabilized and was recorded as $[SO_2]_{Final}$. The time scale for the stabilization of $[SO_2]$ was around 4 min. The OH exposure at each condition is calculated using Eq. (A1):

$$OH\ exposure = \frac{1}{k_{OH,SO_2}} \times -ln\left(\frac{[SO_2]Final}{[SO_2]Initial}\right) \tag{A1}$$

where $k_{OH,SO2} = 9 \times 10^{-13}$ cm$^3$ molec$^{-1}$ is the bimolecular rate constant between OH and $SO_2$ (Davis et al., 1979). The equation above is the result of integrating the differential rate equation for $SO_2$ and assuming pseudo-first order kinetics. The estimation of external OH reactivity (i.e., the OH reactivity with VOCs) requires VOC analysis and is not available in our study. Therefore, the OH exposure shown in this study may have been underestimated.

**Text S2.** Estimation of the mass hygroscopic growth factor (GF) by AIOMFAC model.

Based on the AIOMFAC model (Zuend et al., 2008), we obtained the weight fraction *w* of water and solutes (i.e., dry particles) in $KNO_3$ and KCl particles, respectively. Then, the GF is estimated as equation A2:

$$GF = \frac{m_{wet}}{m_{dry}} = \frac{m_{dry} + m_{water}}{m_{dry}} = 1 + \frac{m_{water}}{m_{dry}} = 1 + \frac{w_{water}}{w_{dry}} \tag{A2}$$

where $m_{wet}$, $m_{dry}$ and $m_{water}$ are the mass of wet particle, dry particle, and particulate water at different relative humidity (RH), respectively. The GF of $KNO_3$ and KCl particles as a function of time is shown in Figure S18. We did not consider efflorescence in the figure since the efflorescence RH for the $KNO_3$ and KCl are around 50%, much lower than 80% used in this study.

**Text S2.** The evolution of organic fragments.

We only showed NF ratios larger than 1 to focus on SOA formation. The positive spectra of both $O_3$-Dark-aged and $O_3$-UV-aged particles show NF increases for +30[NO] or [$CH_2NH_2$] (possibly due to nitrates, oxidized NOC, or amine), +44[$CO_2$] or [$N_2O$] (oxidized organics), +53[$C_4H_5$] and +69[$C_5H_9$] (aromatic hydrocarbons) (Wang et al., 2009; Silva et al., 2000; Dall'osto et al., 2013) (Figure S19). The negative spectra show increases for -137[$C_8H_9O_2$] (possibly methyl guaiacol) (Pagels et al., 2013; Gaie-Levrel et al., 2012) and -57[$C_2HO_2$] (a glyoxylate fragment) (Sullivan et al., 2007; Cheng et al., 2017), as well as -16[O]. $O_3$-UV-aged particles showed 200-folds and 10-folds NF increases for -137[$C_8H_9O_2$] and -57[$C_2HO_2$], respectively, significantly greater than that for UV-aged particles (~2-folds). Compared to OH chemistry, the UV photoactivity of compounds in particulates contributes minorly to organic chemistry (Figure S19).

Figure S20 show the NF of the new peaks in aged particles, which cannot (Dall'osto et al., 2009) be shown in the NF ratio plot due to their absence in fresh particles (zero denominator). $O_3$-Dark-aged particles show NF decreases of m/z +186 to +189 (probably PAHs) (Dall'osto et al., 2009) with increasing [$O_3$]. In contrast, $O_3$-UV-aged particles show NF increases of -31[$CH_3O$] or [HON], +123[$C_7H_7O_2$] and +124[$C_7H_8O_2$] (probably guaiacol) (Diab et al., 2015), and m/z +140 (probably HULIS) (Qin et al., 2006) with increasing OH exposure. These apparent changes in NF of organics fragments indicate the oxidative evolution of organics and likely formation of SOA, although the molecular characterization was hindered by severe fragmentation.

[Figure]

**Figure S1.** The schematic of the experimental set-up. The NOx analyzer and the VOC sensor were used only in the experiment for determining the gas removal efficiency and NOx decay under OH exposure.

[Figure]

**Figure S2.** The appearance of the incense sticks.

[Figure]

**Figure S3.** The emission spectrum of the UVC lamps.

[Figure]

**Figure S4.** RPA of summed nitrate peaks of aged incense burning particles in the presence and absence of charcoal absorber.

[Figure]

**Figure S5.** NF of oxalate and malonate of $O_3$-UV-aged incense burning particles (1500ppb $O_3$ + UV) in the presence and absence of a charcoal absorber.

[Figure]

**Figure S6.** Positive (K- and OC-) and negative spectra (-ON, -ONEC, -Cl, -N, and -ONN) of different categorizations of particles. The spectral characteristics of these categories are similar under different conditions.

[Figure]

**Figure S7.** APA of ON as a function of APA of total nitrate in fresh and $O_3$-Dark-aged particles.

[Figure]

**Figure S8.** $[NO_2^-]/[NO_3^-]$ in water-extract of aged particles. Noted that the $NO_2^-$ in 800ppb $O_3$+UV and 1500ppb $O_3$+UV experiments are undetectable.

[Figure]

**Figure S9.** RPA of total nitrate as a function of RPA of formate.

[Figure]

**Figure S10.** [Formate]/[K$^+$] as a function of [NO$_3^-$]/[K$^+$] in the water-extract of fresh and O$_3$-Dark-aged incense burning particles.

[Figure]

**Figure S11**. The whisker-box plot of the [K$^+$] in the particle water-extract measured by IC normalized by the total counts of collected particles for IC measurement. The error bar shows one standard deviation. We assume there was no new particle formation under ozone exposure since the WCPC showed comparable particle number concentration in the presence and absence of ozone.

[Figure]

**Figure S12.** (a) The difference (aged minus fresh) of the average organic spectra of UV-aged particles; (b) The NF ratio (aged to fresh) of UV-aged particles to fresh particles, as a function of m/z [-150, 150].

[Figure]

**Figure S13.** Relation between the particle collection efficiency of Polystyrene Latex particles and diameter in SPAMS.

[Figure]

**Figure S14.** Evolution of [NOx] under different OH exposure as a function of time. The shadings show one standard deviation. The [NOx] was equilibrated without $O_3$ and UV at 0 min.

[Figure]

**Figure S15.** The NF ratios (aged to fresh) of oxalate and malonate, and TOC ratios as a function of [$O_3$] in $O_3$+UV aging experiments.

[Figure]

**Figure S16.** The total organic content (TOC) ratios of aged to fresh particles extract.

[Figure]

**Figure S17.** Size distribution (0.2-2 μm) of $O_3$-UV-aged particles (at 1500 ppb $O_3$ and UV) and their dicarboxylate-containing fractions. The shadings show one standard deviation.

[Figure]

**Figure S18.** The GF of KCl and $KNO_3$ particles as a function of RH. The red line denotes 80% RH.

[Figure]

**Figure S19.** The NF ratio (aged to fresh) of (a) $O_3$-Dark-aged, (b) $O_3$-UV-aged particles to fresh particles, as a function of m/z [-150, 150].

[Figure]

**Figure S20.** The NF of new organic peaks in aged particles.

**Table S1.** The classification of particles and RPA of total nitrate peaks of aged particles at different laser fluences.

| Experimental Conditions | Laser fluence (mJ) | OC-ON (%) | OC-N (%) | K-ON (%) | K-ONN (%) | K-N (%) | K-ONEC (%) | K-Cl (%) | $RPA_{Total\ nitrate}$ |
|---|---|---|---|---|---|---|---|---|---|
| $O_3$-dark-aged (800 ppb) | 0.6 | (7.6±1.0) | (21.9±1.7) | (29.0±0.7) | (11.2±0.9) | (2.5±0.5) | (22.8±1.8) | (3.9±0.4) | (0.40±0.15) |
| | 1.2 | (7.2±1.4) | (22.6±4.0) | (28.5±4.3) | (10.6±0.2) | (3.3±1.3) | (22.8±2.5) | (4.0±0.1) | (0.37±0.09) |
| $O_3$-UV-aged (800 ppb) | 0.6 | (0.0±0.0) | (36.7±7.2) | (0.0±0.0) | (26.7±2.1) | (35.5±4.2) | (0.0±0.0) | (0.0±0.0) | (0.66±0.03) |
| | 1.2 | (0.0±0.0) | (35.6±2.3) | (0.0±1.0) | (28.6±1.9) | (33.6±4.7) | (0.2±0.2) | (0.0±0.0) | (0.67±0.07) |

**Table S2.** Potential peaks from the inorganics and elemental carbons.

| | m/z | Formula | Ref |
|---|---|---|---|
| Inorganic salts | -163 | $KNO_3$ | (Pratt et al., 2011) |
| | -155 | $Na_2Cl_3$ | (Harrison et al., 2012) |
| | -153 | | |
| | -151 | | |
| | -147 | $Na(NO_3)_2$ | (Ault et al., 2014) |
| | -131 | $NaNO_3NO_2$ | (Ault et al., 2014) |
| | -125 | $H(NO_3)_2$ | |
| | -113 | $KCl_2$ | (Bi et al., 2011) |
| | -111 | | |
| | -109 | | |
| | -101 | $KNO_3/NaNONO_2$ | (Ault et al., 2014) |
| | -97 | $HSO_4/NaCl_2$ | (Liang et al., 2022) |
| | -96 | $SO_4$ | |
| | -95 | $NaCl_2$ | (Ault et al., 2014) |
| | -93 | | |
| | -62 | $NO_3$ | (Cheng et al., 2017) |
| | -58 | NaCl | (Zhu et al., 2020) |
| | -46 | $NO_2$ | (Cheng et al., 2017) |
| | -37 | Cl | (Ault et al., 2014) |
| | -35 | | |
| | +23 | Na | (Dall'osto et al., 2004) |
| | +39 | K | (Bi et al., 2011) |
| | +41 | | |
| | +63 | $Na_2OH$ | (Yang et al., 2009) |
| | +81 | $Na_2Cl$ | (Dall'osto et al., 2004) |
| | +83 | | |
| | +97 | NaKCl | (Gross et al., 2000) |
| | +113 | $K_2Cl$ | (Silva et al., 1999) |
| | +115 | | |

| | +139 | Na₃Cl₂ | (Dall'osto et al., 2004) |
|---|---|---|---|
| | +141 | | |
| Elemental carbons | $\pm 12n$ | $C_n$ | (Zhou et al., 2022) |

**Reference**

Ault, A. P., Guasco, T. L., Baltrusaitis, J., Ryder, O. S., Trueblood, J. V., Collins, D. B., Ruppel, M. J., Cuadra-Rodriguez, L. A., Prather, K. A., and Grassian, V. H.: Heterogeneous Reactivity of Nitric Acid with Nascent Sea Spray Aerosol: Large Differences Observed between and within Individual Particles, The Journal of Physical Chemistry Letters, 5, 2493-2500, 10.1021/jz5008802, 2014.

Bi, X., Zhang, G., Li, L., Wang, X., Li, M., Sheng, G., Fu, J., and Zhou, Z.: Mixing state of biomass burning particles by single particle aerosol mass spectrometer in the urban area of PRD, China, Atmospheric Environment, 45, 3447-3453, https://doi.org/10.1016/j.atmosenv.2011.03.034, 2011.

Cheng, C., Li, M., Chan, C. K., Tong, H., Chen, C., Chen, D., Wu, D., Li, L., Wu, C., and Cheng, P.: Mixing state of oxalic acid containing particles in the rural area of Pearl River Delta, China: implications for the formation mechanism of oxalic acid, Atmospheric Chemistry and Physics, 17, 9519-9533, 2017.

Dall'Osto, M., Harrison, R. M., Coe, H., and Williams, P.: Real-time secondary aerosol formation during a fog event in London, Atmos. Chem. Phys., 9, 2459-2469, 10.5194/acp-9-2459-2009, 2009.

Dall'Osto, M., Beddows, D. C. S., Kinnersley, R. P., Harrison, R. M., Donovan, R. J., and Heal, M. R.: Characterization of individual airborne particles by using aerosol time-of-flight mass spectrometry at Mace Head, Ireland, Journal of Geophysical Research: Atmospheres, 109, https://doi.org/10.1029/2004JD004747, 2004.

Dall'Osto, M., Ovadnevaite, J., Ceburnis, D., Martin, D., Healy, R. M., O'Connor, I. P., Kourtchev, I., Sodeau, J. R., Wenger, J. C., and O'Dowd, C.: Characterization of urban aerosol in Cork city (Ireland) using aerosol mass spectrometry, Atmospheric Chemistry and Physics, 13, 4997-5015, 2013.

Davis, D., Ravishankara, A., and Fischer, S.: SO2 oxidation via the hydroxyl radical: atmospheric fate of HSOx radicals, Geophysical Research Letters, 6, 113-116, 1979.

Diab, J., Streibel, T., Cavalli, F., Lee, S., Saathoff, H., Mamakos, A., Chow, J., Chen, L.-W., Watson, J., and Sippula, O.: Hyphenation of a EC/OC thermal–optical carbon analyzer to photo-ionization time-of-flight mass spectrometry: an off-line aerosol mass spectrometric approach for characterization of primary and secondary particulate matter, Atmospheric Measurement Techniques, 8, 3337-3353, 2015.

Gaie-Levrel, F., Perrier, S., Perraudin, E., Stoll, C., Grand, N., and Schwell, M.: Development and characterization of a single particle laser ablation mass spectrometer (SPLAM) for organic aerosol studies, Atmospheric Measurement Techniques, 5, 225-241, 2012.

Gross, D. S., Gälli, M. E., Silva, P. J., and Prather, K. A.: Relative sensitivity factors for alkali metal and ammonium cations in single-particle aerosol time-of-flight mass spectra, Analytical Chemistry, 72, 416-422, 2000.

Harrison, R. M., Dall'Osto, M., Beddows, D., Thorpe, A. J., Bloss, W. J., Allan, J. D., Coe, H., Dorsey, J. R., Gallagher, M., and Martin, C.: Atmospheric chemistry and physics in the atmosphere of a developed megacity (London): an overview of the REPARTEE experiment and its conclusions, Atmospheric Chemistry and Physics, 12, 3065-3114, 2012.

Liang, Z., Zhou, L., Infante Cuevas, R. A., Li, X., Cheng, C., Li, M., Tang, R., Zhang, R., Lee, P. K. H., Lai, A. C. K., and Chan, C. K.: Sulfate Formation in Incense Burning Particles: A Single-Particle Mass Spectrometric Study, Environmental Science & Technology Letters, 10.1021/acs.estlett.2c00492, 2022.

Pagels, J., Dutcher, D. D., Stolzenburg, M. R., McMurry, P. H., Gälli, M. E., and Gross, D. S.: Fine-particle emissions from solid biofuel combustion studied with single-particle mass spectrometry: Identification of markers for organics, soot, and ash components, Journal of Geophysical Research: Atmospheres, 118, 859-870, https://doi.org/10.1029/2012JD018389, 2013.

Pratt, K., Murphy, S., Subramanian, R., DeMott, P., Kok, G., Campos, T., Rogers, D., Prenni, A., Heymsfield, A., and Seinfeld, J.: Flight-based chemical characterization of biomass burning aerosols within two prescribed burn smoke plumes, Atmospheric Chemistry and Physics, 11, 12549-12565, 2011.

Qin, X. and Prather, K. A.: Impact of biomass emissions on particle chemistry during the California Regional Particulate Air Quality Study, International Journal of Mass Spectrometry, 258, 142-150, https://doi.org/10.1016/j.ijms.2006.09.004, 2006.

Silva, P. J. and Prather, K. A.: Interpretation of mass spectra from organic compounds in aerosol time-of-flight mass spectrometry, Analytical Chemistry, 72, 3553-3562, 2000.

Silva, P. J., Liu, D.-Y., Noble, C. A., and Prather, K. A.: Size and Chemical Characterization of Individual Particles Resulting from Biomass Burning of Local Southern California Species, Environmental Science & Technology, 33, 3068-3076, 10.1021/es980544p, 1999.

Sullivan, R. C. and Prather, K. A.: Investigations of the diurnal cycle and mixing state of oxalic acid in individual particles in Asian aerosol outflow, Environmental Science Technology, 41, 8062-8069, 2007.

Wang, X., Zhang, Y., Chen, H., Yang, X., Chen, J., and Geng, F.: Particulate nitrate formation in a highly polluted urban area: a case study by single-particle mass spectrometry in Shanghai, Environmental Science & Technology, 43, 3061-3066, 2009.

Yang, F., Chen, H., Wang, X., Yang, X., Du, J., and Chen, J.: Single particle mass spectrometry of oxalic acid in ambient aerosols in Shanghai: Mixing state and formation mechanism, Atmospheric Environment, 43, 3876-3882, 2009.

Zhou, L., Li, M., Cheng, C., Zhou, Z., Nian, H., Tang, R., and Chan, C. K.: Real-time chemical characterization of single ambient particles at a port city in Chinese domestic emission control area — Impacts of ship emissions on urban air quality, Science of The Total Environment, 819, 153117, https://doi.org/10.1016/j.scitotenv.2022.153117, 2022.

Zhu, S., Li, L., Wang, S., Li, M., Liu, Y., Lu, X., Chen, H., Wang, L., Chen, J., and Zhou, Z.: Development of an automatic linear calibration method for high-resolution single-particle mass spectrometry: improved chemical species identification for atmospheric aerosols, Atmospheric Measurement Techniques, 13, 4111-4121, 2020.

Zuend, A., Marcolli, C., Luo, B. P., and Peter, T.: A thermodynamic model of mixed organic-inorganic aerosols to predict activity coefficients, Atmos. Chem. Phys., 8, 4559-4593, 10.5194/acp-8-4559-2008, 2008.